# Development of an ObLiGaRe Doxycycline Inducible Cas9 system for pre-clinical cancer drug discovery

Anders Lundin [1,10], Michelle J. Porritt [1,10], Himjyot Jaiswal[1,8], Frank Seeliger[2], Camilla Johansson[3], Abdel Wahad Bidar[3], Lukas Badertscher[1], Sandra Wimberger[1], Emma J. Davies[4,9], Elizabeth Hardaker[4], Carla P. Martins[4], Emily James[4], Therese Admyre[1], Amir Taheri-Ghahfarokhi [1], Jenna Bradley[5], Anna Schantz[6], Babak Alaeimahabadi[7], Maryam Clausen[1], Xiufeng Xu[1], Lorenz M. Mayr[1], Roberto Nitsch [1], Mohammad Bohlooly-Y[1], Simon T. Barry[4] & Marcello Maresca [1✉]

The CRISPR-Cas9 system has increased the speed and precision of genetic editing in cells and animals. However, model generation for drug development is still expensive and time-consuming, demanding more target flexibility and faster turnaround times with high reproducibility. The generation of a tightly controlled ObLiGaRe doxycycline inducible SpCas9 (ODInCas9) transgene and its use in targeted ObLiGaRe results in functional integration into both human and mouse cells culminating in the generation of the ODInCas9 mouse. Genomic editing can be performed in cells of various tissue origins without any detectable gene editing in the absence of doxycycline. Somatic in vivo editing can model non-small cell lung cancer (NSCLC) adenocarcinomas, enabling treatment studies to validate the efficacy of candidate drugs. The ODInCas9 mouse allows robust and tunable genome editing granting flexibility, speed and uniformity at less cost, leading to high throughput and practical preclinical in vivo therapeutic testing.

[1] Translational Genomics, Discovery Sciences, BioPharmaceuticals R&D, AstraZeneca, Gothenburg, Sweden. [2] Clinical Pharmacology and Safety Sciences, BioPharmaceuticals R&D, AstraZeneca, Gothenburg, Sweden. [3] Clinical Pharmacology and Safety Sciences, Sweden Imaging Hub, BioPharmaceuticals R&D, AstraZeneca, Gothenburg, Sweden. [4] Early Oncology TDE, Oncology R&D, AstraZeneca, Li KaShing Centre, Cambridge, UK. [5] Discovery Sciences, BioPharmaceuticals R&D, AstraZeneca, Cambridge Science Park, Cambridge, UK. [6] Pharmaceutical Sciences, Discovery Sciences, BioPharmaceuticals R&D, AstraZeneca, Gothenburg, Sweden. [7] Data Sciences and Quantitative Biology, Discovery Sciences, BioPharmaceuticals R&D, AstraZeneca, Gothenburg, Sweden. [8] Present address: Cellink AB, Gothenburg, Sweden. [9] Present address: Healx, Cambridge, UK. [10] These authors contributed equally: Anders Lundin, Michelle J. Porritt. ✉email: Marcello.Maresca@astrazeneca.com

Genetic manipulations in cells and organisms are used to understand the role of genes in a physiological or disease context. One strategy has been to exploit nucleases such as zinc finger nucleases (ZFNs), transcription activator-like effector nucleases, and clustered regularly interspaced short palindromic repeats (CRISPR) for targeted insertions and deletions (indels) or precise genome edits[1–4]. Among those nucleases, the CRISPR-Cas9 system has revolutionized precision molecular genetic approaches, accelerating the generation of genetically engineered cells and animal models (reviewed in Adli[5]).

Genetic manipulation of mice to create disease models or validate therapeutic targets are fundamental to most areas of biology. In oncology research, there is significant investment in genetically engineered mouse models (GEMMs) where gene expression is both spatially and temporally controlled to enable more faithful modeling of equivalent human tumors. While GEMMs are powerful tools, mouse strain production is both time-consuming and costly and needs to be repeated for each new engineering. Moreover, GEMMs are expensive to maintain with each specific model requiring extensive breeding to maintain a single model. A further requirement of somatic multiplex-mutagenesis is extensive mouse intercrossing for the generation of relevant experimental cohorts of multi-allelic mutant mice[6]. In addition to long breeding programs, the tumor models have a long indolence time, which often precludes the use of these models for drug testing[7]. When these models have a latency period conducive to drug testing, due to the complex nature of breeding programs required, often only small numbers of tumor bearing animals can be recruited onto study at a given time, compared to more traditional xenografted or explant cancer models[8]. To reduce cost and increase output, it is preferential to have tunable timelines to adapt tumor development to treatment time based on compound activity and mode of action[6].

In contrast, non-germline GEMMs (nGEMMS), where somatic cells carry engineered alleles but not germline cells, provide an option to shorten production time lines and increase genomic heterogeneity[9]. However, timelines to establishment of tumor models are still long, up to months before having tumors in which to test therapeutics (reviewed in Heyer et al.[10]). Moreover, traditional xenograft nGEMMs are also often immunodeficient, limiting applicability[6]. Furthermore, cancer prone non-germline chimeric mice, produced by injecting genetically engineered embryonic stem cells into blastocysts of a mouse line, have an increased variability related to tumorigenesis between individual mice[11]. This negatively affects standardizing treatment studies in these models.

Instead, nGEMMs, which rely on in vivo delivery of editing components to generate somatic genomic changes of oncogenes and/or repressor genes, provide greater flexibility in target turn-over and recapitulation of tumor microenvironment, including immune effects. Gene manipulation is often achieved by viral delivery, using either an adeno-associated virus, adenovirus, or lentivirus[12,13]. Due to limitations in the viral packaging capacity, nucleases, target sequences, and repair templates are often separated into different vectors, resulting in lower editing efficacy[14]. The introduction of constitutive expressing Cas9 mice, Rosa26-Cas9, reduced the number of components that needed to be delivered to the cell thereby increasing editing efficacy[13,14].

The utility of constitutive Cas9 systems is however limited by off-target effects and inflammatory responses if Cas9 expression is not controlled. Moreover recent data indicate that the constitutive expression of Cas9 upregulates a Trp53 response[15].

Inducible expression systems including recombinase-mediated cassette exchange, Cre-recombinase, or tetracycline (Tet) sensitive systems, have been used to generate inducible Cas9 mice[12,13,16–21].

Cre systems are limited by potential genotoxicity due to cryptic recombination sites in mammalian genomes[22], which is not observed with doxycycline (dox) or Tet-inducible systems. Moreover, a Cre-based system allows only for a single activation followed by constitutive expression while a Tet-inducible system can repeatedly be turned on and off, thereby avoiding downstream effects of constitutive expressions. However, tight regulation of the Tet system is challenging so common practice is to split the two major components (Tet repressor and inducible promotor) and introduce the two different parts in different loci, thus complicating the genetic engineering process[23].

Here, we describe the generation a transgene, Obligate Ligation-Gated Recombination (ObLiGaRe) Doxycycline Inducible Cas9 (ODInCas9), an all in one, universal Tet-On system where a combination of insulators placed in a modular vector allows tight temporally regulated expression of *Streptococcus pyogenes* Cas9 (SpCas9). The ODInCas9 transgene carries human and mouse ZFN sites for targeted ObLiGaRe-mediated integration into both human and mouse genomes. ObLiGaRe is the first described knock in method of targeted integration by NHEJ that is using nuclease-mediated cleavage of the donor vector and the genomic target locus followed by endogenous NHEJ-mediated ligation. ObLiGaRe allows efficient and directional integration of monomer or ligated multimers of a cassette of interest at a specific locus. The use of NHEJ renders the integration mechanism independent from cell cycle status and the size of the cassette, giving more flexibility respect to homology-dependent systems of targeted integration. In this work we use ObLiGaRe to enable the generation of Cas9-inducible human iPSCs and mouse ESCs. ODInCas9 is functional and tightly controlled in vitro and in vivo. The ODInCas9 mice demonstrate inducible Cas9 expression in all tissues and can be repeatedly induced with no Cas9 detection upon dox withdrawal. Delivery of sgRNAs as a single component to Cas9 expressing cells demonstrates high editing efficacy resulting in disease phenotype readouts with no detectable edits in the absence of dox. For oncology research, the ODInCas9 cassette offers the ability to introduce multiple sequential genetic alterations and better replicate the natural mutation patterns accumulated in tumors with time. Further, delivery of single-stranded DNA template for increased homologous directed repair can help establish a more precise oncogene-driven cancer model. The ODInCas9 mouse allows tissue-specific induction of tumors in a relevant niche that can develop and progress to late stage disease much like human tumors. The ODInCas9 mouse model provides a robust and tunable induction of tumors allowing for flexibility, speed, and uniformity at reduced cost, leading to high throughput and practical preclinical in vivo therapeutic testing models.

## Results

**ODInCas9 transgene expression in multiple cellular backgrounds.** To achieve inducible Cas9 expression in human and mouse cell lines of various tissue origins, the ODInCas9 transgene cassette was developed. Inducible Cas9 expression by a tetracycline response element promoter (TRE3G) provides simple and fast genomic engineering. Inverted ZFN sites targeting Rosa26 or AAVS1 of mouse or human genomes, respectively, facilitate targeted transgene integration using ObLiGaRe[3]. β-globulin insulators downstream of Tet-On 3G, which flanks TREG3-Cas9-T2A-GFP, allow for the generation of an all-in-one system without expression in the inactive state (Fig. 1a, b).

Integration of the ODInCas9 transgene into multiple human cancer cell lines, originating from organs such as ovary (OVCAR8), colon (HCT116), kidney (HEK293), liver (HepG2), lung (A549), mouse brain cells (N2a), as well as hiPSC (Supplementary Fig. 1a, b), confirmed universal integration. The ODInCas9 transgene offers

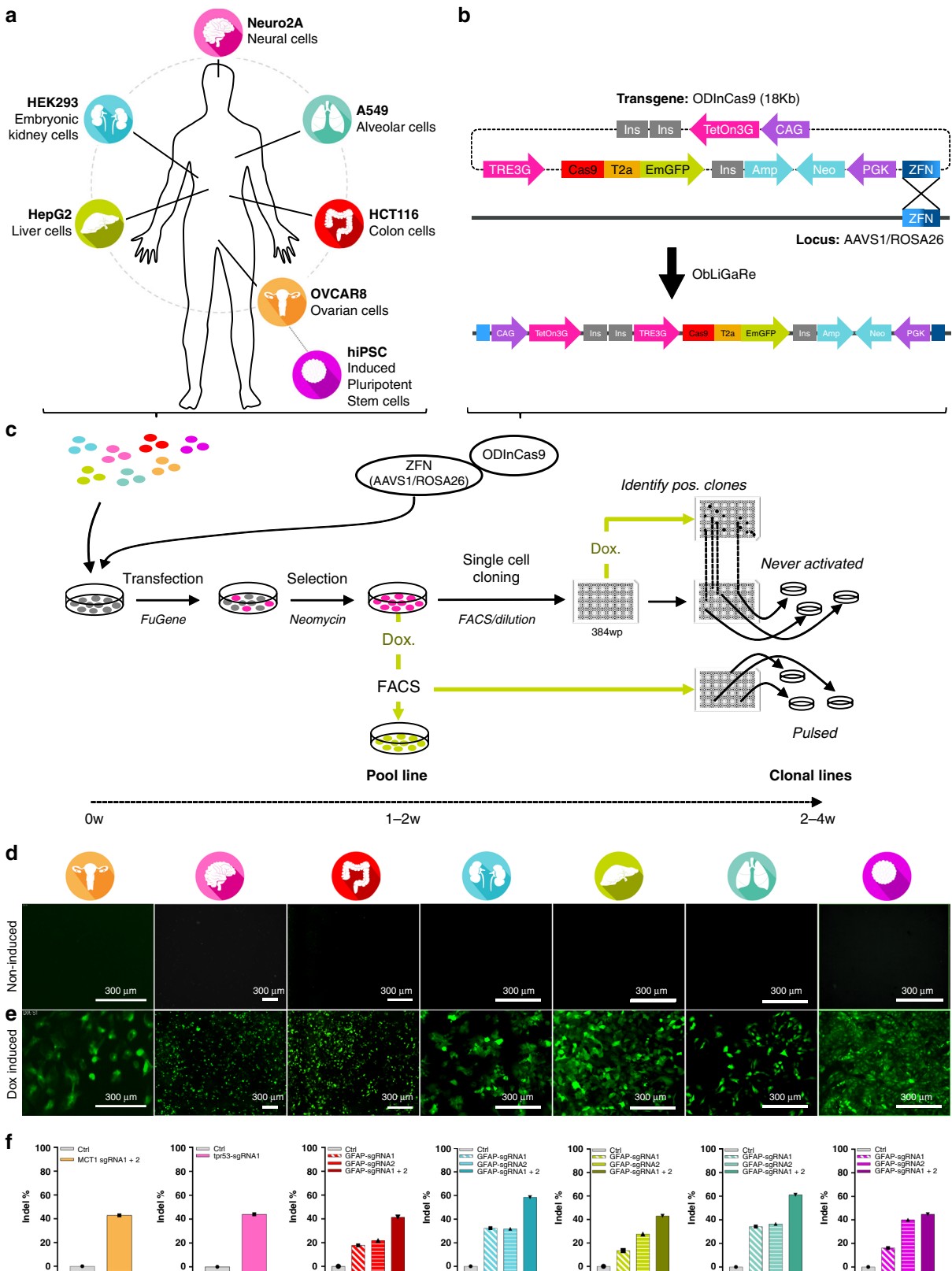

**Fig. 1 ObLiGaRe Dox Inducible (ODIn) Cas9, an all in one, universal TET ON inducible system.** ObLiGaRe Dox-inducible (ODIn) Cas9, an all-in-one, universal Tet-On inducible system **a** human and mouse cell lines sourced from brain (N2A), lung (A549), colon (HCT116), ovarian tissue (OVCAR8), liver (HepG2), kidney (HEK293), and human-induced pluripotent stem cells (iPSC) engineered using **b** the ODInCas9 system targeting the AAVS1 or ROSA26 locus using ZFN-directed ObLiGaRe. **c** Outlined methodology for rapid cell line generation of ODInCas9 pools and clonal lines encountered with or without exposure to Cas9. Evaluation of GFP expression of **d** non-induced and **e** Doxycycline Inducible ODInCas9 cell lines. **f** Quantification of mismatch endonuclease assay estimating indel formation post sgRNAs transfection of induced ODInCas9 cell lines targeting; MCT1 in OVCAR8, Tpr53 in N2A, and GFAP in HCT116, HEK293, HepG2, A549, and iPSC ($n = 7$ biological independent cell lines). Scale bar: 300 μm.

the ability to select clones through neomycin selection and GFP fluorescent sorting to generate cell lines within 2–6 weeks (Fig. 1c). Dox treatment of ODInCas9 cell lines induced strong system activation, detectable by GFP expression. However, no activation (GFP signal) was observed in the absence of dox (Fig. 1d, e). Importantly, normal karyotype, pluripotency, and differentiation capacity was maintained (Supplementary Fig. 1d, f, g) in hiPSC with ODInCas9 transgene integration (Supplementary Fig. 1c, f). Upon activation, ODInCas9 can generate genomic edits in hiPSC, HCT116, HEK293, HepG2, A549, OVCAR8, and N2a cells using either a single sgRNA or paired sgRNAs, (Fig. 1f), which by NGS analysis showed over 85% editing efficiency in hiPSCs (Supplementary Fig. 1i). Together, this demonstrates that ODInCas9 transgene is functional in multiple cellular backgrounds of both human and mouse origin.

**Dose- and time-dependent control of the ODInCas9 transgene enables high editing efficiency**. ODInCas9 transgene expression is tightly controlled and both transcript and protein levels correlate to dox concentration and treatment times. Dose response analysis showed detectable Cas9 and GFP expression from dox induction as low as 1–5 ng/ml in both hiPSC (Fig. 2a–c) and cancer cell lines (Supplementary Fig. 2a–d). Flow cytometry analysis showed a uniform expression of GFP in ODInCas9 hiPSC (Fig. 2d) and both human and mouse ODInCas9 cell lines (Supplementary Fig. 2e). Expression levels correlated with transgene copy number integration, as the hiPSC ODInCas9 homozygous clonal line displayed two times the level of GFP intensity and Cas9 protein expression compared to the heterozygous clone (Supplementary Fig. 2f–h). Continuous stimulation with dox resulted in stable expression over time. Robust expression was detected after 6-h induction (Fig. 2e, f). Moreover, a 1 h dox treatment demonstrated transient GFP and Cas9 expression going down to undetectable levels after 4–5 days (Fig. 2g–i and Supplementary Fig. 2i). In summary, the ODInCas9 transgene is highly sensitive across multiple cellular backgrounds demonstrating both the flexibility of stable and transient expression.

The broad editing potential was demonstrated by liposomal delivery of a sgRNA targeting repetitive Alu sequences in the human genome. This revealed efficient editing causing cytotoxicity due to multitude of double-stranded breaks resulting in significant reduction in both proliferation and viability (Supplementary Fig. 2j, k). Liposomal and lipid nanoparticle (LNP) delivery vehicles in combination with plasmid, full length, and two-component synthetic sgRNA showed targeted genomic editing (Fig. 2j–l and Supplementary Fig. 2l–n). Synthetic crRNA sequence information is within Table 1 and Supplementary Table 1. Assessing downstream effects of genomic editing guides individually targeting the surface protein GPI anchor PIGM, transmembrane protein MCT1, and essential gene CDK12 were delivered either by liposomal reagent or electroporation. Using different guides, PIGM was efficiently edited with 90% protein KO, determined by fluorescent analysis of PIGM, which had been tagged using a fluorescently labeled aerolysin variant (Fig. 2m, n). Similarly, targeting CDK12, a gene involved in DNA damage response, resulted in significant reduction of cell proliferation upon protein knockdown in contrast to reduction of the nonessential transmembrane protein MCT1 (Supplementary Fig. 2k, o, p). Together, ODInCas9 transgene induction in combination with standard transfection techniques of regular sgRNA components demonstrate highly efficient editing, with significant downstream effects on protein expression and cell biology (primer sequences are in Table 1 and Supplementary Table 2).

Expression of the ODInCas9 transgene in unstimulated conditions was below the threshold of detection by different techniques. Cas9 transcript levels in inactivated hiPSC ODInCas9 were comparable to *TBX1*, *SOX17*, and *SOX1* (Supplementary Fig. 2q), biomarkers of germline differentiation, not having protein translation in pluripotent hiPSC. Dox induction significantly changed *Cas9* transcript expression (Supplementary Fig. 2r) to levels of *OCT4* (Supplementary Fig. 2q), a highly expressed gene (Supplementary Fig. 1f) driving pluripotency. Non-induced conditions have no detectable Cas9 protein or GFP intensity (Fig. 2a–i) analyzed by flow cytometry (Fig. 2d and Supplementary Fig. 2i, j). Delivery of sgRNA targeting repetitive Alu elements to non-induced ODInCas9 hiPSC do not affect proliferation (Supplementary Fig. 2j) and no genomic editing can be observed for delivery of sgRNA targeting specific genomic sites (Fig. 2j–l). This demonstrates that expression of the ODInCas9 transgene is prevented in the absence of dox.

**Repeatable controlled induction of Cas9 expression**. The ODInCas9 remains inactive after transient activation. Targeting intermediate filament, GFAP, and E3 ubiquitin-protein ligase, MYLIP, respectively, by repeating 1 h pulse inductions at two time points, days 0 and 4, (Fig. 2o), generated edits at each respective locus (d4:1i, d7:1ii) (Fig. 2p, q and Supplementary Fig. 2s, t) while being turned off in between editing events (Fig. 2r and Supplementary Fig. 2u). No editing could be detected in the completely inactive control (d4:1, d7:1), but more importantly, the condition carrying a GFAP edit from the first induction (d4:1i) showed no edit formation at MYLIP upon repeated transfection in the inactive state (d7:1i) (Fig. 2q). All together this demonstrates that the ODInCas9 transgene is highly sensitive and versatile, additionally allowing the introduction of sequential edits.

**ODInCas9 mouse**. The highly controlled features of the ODInCas9 cassette make it ideal to drive transient induction of Cas9 expression in vivo. As Cas9 expression is non-detectable or below the threshold of detection in the uninduced state and returns to baseline level following transient induction, it provides the ideal cassette to generate a transgenic mouse without the drawbacks associated with leaky or sustained expression of Cas9. To generate a construct suitable for generation of transgenic mice the ODInCas9 transgene was inserted into the *Rosa26* locus using Rosa26 ZFN-mediated ObLiGaRe[3]. Validation of transgene induction in mESC ODInCas9 cells was performed, with both GFP fluorescent signal and Cas9 protein expression observed (Fig. 3a, b). The validated clone was then used to generate chimeric mice which were then expanded by breeding to a C57Bl/6NCrl background. Consistent with the tight regulation of the Cas9 expression seen ex vivo, heterozygote mice do not display any phenotype unless dosed with dox. Moreover, the classic mendelian inheritance observed when bred to C57BL/6NCrl mice (Fig. 3c) suggest no background toxicity from the construct. Homozygous mice are fertile and do not display any visible phenotype in the absence of dox.

**Ubiquitous distribution of Cas9 protein in the ODInCas9 mouse**. To induce Cas9 expression, animals were provided drinking water with 2 mg/ml dox. Following dox induction, heterozygous ODInCas9 mice expressed Cas9 protein in all tissues, consistent with the expression pattern previously reported in the Rosa26 locus[24]. Dox treatment in heterozygous mice for 24 h results in intense staining of the GI tract and pancreas with the lowest expression observed in skeletal muscle (Fig. 3d). Seventy-two hours of dox treatment increase the number of organs with high Cas9 and GFP expression; lung, liver, kidney, heart, and skeletal muscle (Fig. 3e, g and Supplementary Fig. 3c). Detection

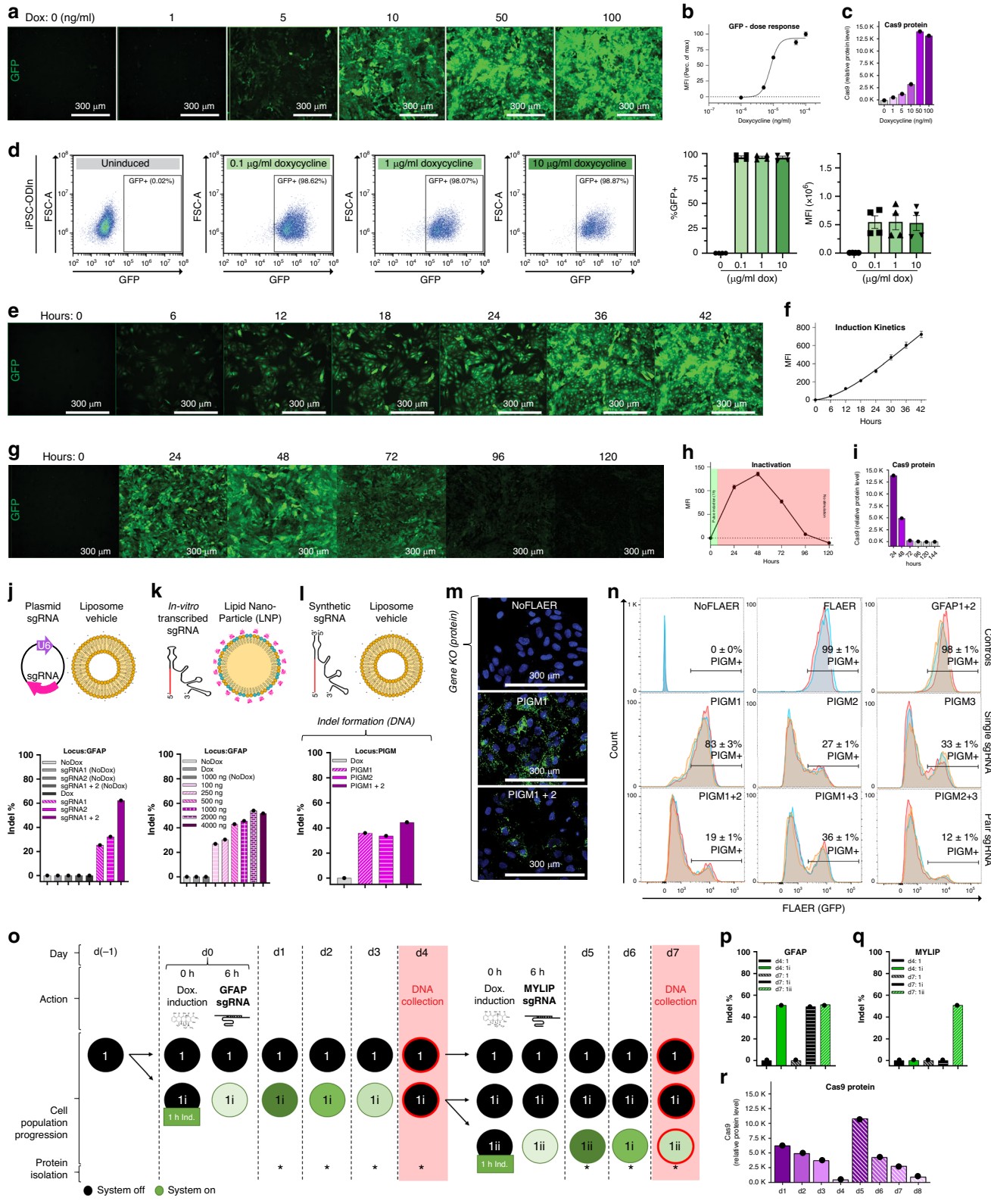

of Cas9 and GFP by western blotting is consistent with microscopic fluorescence (Fig. 3d, e, g and Supplementary Fig. 3a, b). Homozygous ODInCas9 mice have greater Cas9 and GFP expression in the liver compared with heterozygous mice (Fig. 3f, h). No Cas9 or GFP was detected in the absence of dox stimulation (Fig. 3f, h). Given that heterozygous mice give robust

expression of Cas9 to maximize animal usage from each breeding, and to minimize toxicity associated with high Cas9 induction, heterozygous mice of both sexes (aged between 10 and 18 weeks) were used in subsequent studies.

Broad expression of Cas9 was achieved in the heterozygous mice. Cellular distribution of Cas9 expression within organs was

**Fig. 2 ODInCas9 system characterization and validation. a** Human iPSC (hiPSC) ODInCas9 C86 GFP expression 48 h post induction start displayed as **b** MFI evaluated by imaging ($n = 9$ per concentration; mean ± SEM; all normalized to maximum fluorescence intensity at 100 ng dox). **c** Cas9 protein expression in response to dox concentration from 0 to 100 ng/ml (normalized to actin concentration, associated to Supplementary Fig. 2c). **d** FACS evaluation of cell population activation at 48 h post dox induction at concentrations 0.1, 1, and 10 µ/ml ($n = 4$ per concentration, mean ± SEM). **e** GFP induction kinetics of hiPSC ODInCas9 C86 induced with 100 ng/ml dox displayed as **f** MFI evaluated by imaging ($n = 9$ per concentration; mean ± SEM). **g** Transgene expression of hiPSC ODInCas9 C86 after 1 h dox treatment (10 µg/ml) following washout evaluated by **h** GFP MFI and **i** Cas9 protein expression (normalized to actin concentration, associated to Supplementary Fig. 2i). Endonuclease mismatch activity assay estimating Cas9 indel formation 48 h post transfection of activated hiPSC ODInCas9 C86 using **j** sgRNA plasmid in combination with liposome vehicle delivery (associated to Supplementary Fig. 2l), **k** in vitro transcribed sgRNA in combination with lipid nanoparticles delivery at concentrations between 100 and 4000 ng (associated to Supplementary Fig. 2m) and **l** synthetic sgRNA with liposome vehicle delivery (associated to Supplementary Fig. 2n). Protein knock-out evaluation of GPI anchor PIGM in hiPSC ODInCas9 C86 treated with sgRNA and paired sgRNA combinations visualized by **m** image representation of FLAER assay staining and **n** FACS analysis of PIGM+ cell population ($n = 2$ for staining controls (NoFLAER and FLAER) and $n = 3$ per locus targeted conditions; mean ± SEM). **o** Experimental design of cellular sub-culturing during repeated 1 h activation of hiPSC ODInCas9 C86 at d0 and d4 with simultaneous treatment of paired sgRNAs targeting GFAP and MYLIP, respectively. Mismatch endonuclease activity assay estimating indel frequency of samples isolated at d4 and d7 at **p** GFAP and (associated to Supplementary Fig. 2s) **q** MYLIP locus sites (associated to Supplementary Fig. 2t). **r** Cas9 protein levels evaluated daily during repeated 1 h activation of hiPSC ODInCas9 C86 (normalized to actin concentration, associated to Supplementary Fig. 2u). * = protein isolation. MFI mean fluorescent intensity, dox doxycycline. Scale bar: 300 µm.

| Table 1 sgRNA and primer sequences used to demonstrate editing efficiency. | | | |
|---|---|---|---|
| **Target** | **Guide RNA sequence** | **Forward primer** | **Reverse primer** |
| GFAP | GGGTGCCAGGACCCAGACGG | TCATCATGGTCCAACCAACC | GAAGCGAACCTTCTCGATGT |
| GFAP | GGAGACCCGGGCCAGTGAGC | | |
| MYLIP | AGGGCAGAAACTGCTCAT | CTCTTGGTGTCCTCCAGCAT | TTGGGCTAAAGTCATCTTTACAA |
| MYLIP | TGGAAAACTATGGCATAGAA | | |
| PIGM | AACTTGAAGGTGGCTCCAGC | AGTCTGCAGTCGTTTCGGTT | TCTATGGTTTCGCGGTGCAT |
| PIGM | TGTCCGTATACCTCACGTGC | | |
| PIGM | TGGATACTGCCATAGGCAGG | | |
| MCT1 | CGTATAGTCATGATTGTTGG | TGTGAGGGAGCAGTTTCCT | GCCAGCCATAAACTAATGCTTC |
| MCT1 | ACAGACGTATAGTTGCTGTA | | |

confirmed by immunohistochemistry (Supplementary Fig. 3e–m and Supplementary Table 3). Cas9 is expressed in lung epithelial cells, hepatic cells in the liver, tubules around the glomerulus in the kidney and in exocrine epithelial cells of the pancreas (Supplementary Fig. 3e–h). The Cas9 positive cells within the seminal vesicles were in the seminiferous epithelium in mucosal folds, epithelial cells of the intestines, skin, and spleen whereas in the brain, staining was limited to ependymal cells of the lateral ventricles (Supplementary Fig. 3i–m).

**Controlled repeated induction of Cas9 in the ODInCas9 mouse.** Treatment of the ODInCas9 mouse with dox induces Cas9 and GFP expression and removal of dox results in loss of Cas9 expression determined via GFP fluorescence (Fig. 3i). Re-introduction of dox treatment, 3 weeks after the first induction, induces Cas9 expression again (Fig. 3i), demonstrating the tightness of the temporal expression of Cas9 in vivo consistent with the observations made in cell lines.

**Targeted editing in the ODInCas9 mouse liver.** To determine whether targeted gene editing in the liver could be achieved in the ODInCas9 mouse, the ODInCas9 mouse was crossed with the human *PCSK9* knock in hypercholesterolemic mouse[25]. The liver was chosen as the target organ model as LNP-coated sgRNAs readily accumulate in the organ. Mice ($n = 18$ PCSK9 Het;ODInCas9 Het and $n = 17$ PCSK9 wild type;ODInCas9 Het) were intravenously (i.v.) dosed with LNP containing the validated sgRNA for *PCSK9* and killed after 21 days. LNP delivery of sgRNA edited ~3.5% reads covering the *PCSK9* gene and had a physiological effect, significantly reducing plasma triglyceride levels by 20% (*T*-test, $t(18) = 2,45$, $p = 0.026$) (Fig. 3j, l, m). No

editing or changes in plasma triglycerides were observed in LNP-treated wild-type mice.

While LNP can deliver sgRNAs to certain tissues, it is not a vehicle capable of delivering sgRNAs to tissues beyond the liver at high efficiencies. Moreover, LNP delivery is associated with inflammation and other toxicities. Virus particles are an alternative vehicle with which to deliver sgRNA with higher efficiencies and greater organ specificity. The adeno-associated virus serotype 9 (AAV9) has broad tissue tropism and is not associated with systemic inflammation. AAV9 with a sgRNA targeting *Trp53* was delivered i.v. to ODInCas9 mice. Dox-treated mice had significant (5%) editing in the liver after 3 days (Fig. 3j, k). In uninduced ODInCas9 mice, AAV9-delivered sgRNA resulted in no editing of the *Trp53* gene (99,9% wild-type expression). sgRNA and primer sequence information is within Supplementary Table 4. This demonstrates that AAV9 delivery of sgRNA to transgenic mice bearing the ODInCas9 transgene has potential to enable targeted CRISPR modulation of multiple tissues.

**Generation of genetically complex tumor models in the ODInCas9 mouse.** The ODInCas9 mouse has significant potential as a universal background in which to rapidly generate stable gene modification of tissues. Modeling tumor biology in mouse models requires tissue-specific mutation, deletion or truncation of multiple genes. Moreover, to understand the impact of a novel gene on tumor development requires time for generation of a new mouse or establishing complex breeding programs. Through conventional routes this requires long breeding programs and can be hampered by long indolence time for tumors to appear. To establish whether the ODInCas9 mouse can be used to induce key

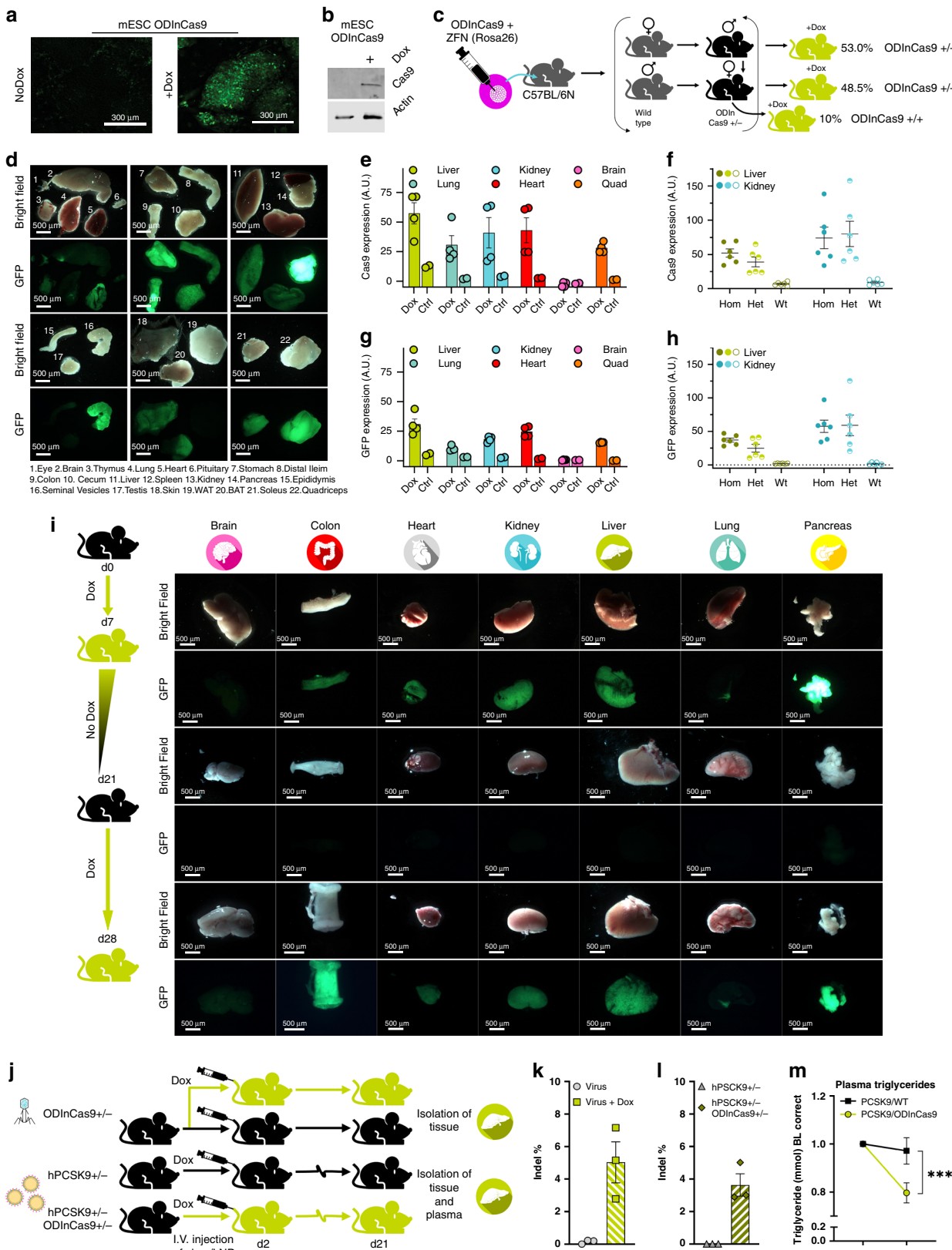

1.Eye 2.Brain 3.Thymus 4.Lung 5.Heart 6.Pituitary 7.Stomach 8.Distal Ileim
9.Colon 10. Cecum 11.Liver 12.Spleen 13.Kidney 14.Pancreas 15.Epididymis
16.Seminal Vesicles 17.Testis 18.Skin 19.WAT 20.BAT 21.Soleus 22.Quadriceps

cancer mutations in a specific organ that leads to tumor formation, we delivered AAV to the lung.

As AAV9 has high trophism in lung tissue and with precedence for transforming lung tissue with inhaled viruses[12], the ability to

generate lung tumors was assessed. To induce lung tumors, AAV9 virus bearing sgRNA to modify the *Kras* oncogene and *Trp53* tumor suppressor were generated. A common strategy was to generate a gain of function point mutation in *Kras* (G12D or G12C)

**Fig. 3 Generation and validation of the ODInCas9 mouse. a** Validation of ODInCas9 mouse ESC GFP and **b** Cas9 expression with and without Dox. **c** Breading strategy of heterozygote and homozygote ODInCas9 mice. **d** Validation of GFP expression in multiple organs 72 h post dox induction by fluorescent imaging. Quantification of WB of **e** Cas9 and **g** GFP protein expression in liver, lung, kidney, heart, brain, and quadricep of heterozygous ODInCas9 mice (normalized to GAPDH concentration, associated to Supplementary Fig. 3c) using $n = 4$ and $n = 2$ animals for dox-induced and control conditions, respectively. Quantification of WB of **f** Cas9 and **h** GFP expression in liver and kidney of heterozygous and homozygous mice (normalized to GAPDH concentration, associated to Supplementary Fig. 3d) using $n = 6$ animals for all groups. GFP intensity **i** in brain, colon, heart, kidney, liver, lung, and pancreas measured at multiple time points; d7, d21, and d28 following 7 days doxycycline stimulation, 14 days washout without dox stimulation, and 7 days of dox stimulation. Schematic of **j** Intravenous injection of AAV targeting *Trp53* in ODInCas9 mice and LNP delivered sgRNA targeting human PCSK9 in ODInCas9 mice crossed with the human knock in PCSK9 mouse following dox or no dox stimulation. Indel analysis of amplicon sequencing of **k** *Trp53* and **l** PCSK9 of isolated liver tissue, $n = 3$ animals per treatment group. **m** Plasma triglycerides decreased from baseline and 21 days post LNP delivered guide in PCSK9;ODInCas9 mice ($t(18) = 2,45$, $p = 0.026$). Experimental set-up included $n = 18$ animals at baseline per treatment group but PCSK9 and PCSK9; ODInCas9 mice had $n = 11$ and $n = 9$ at end point assessment due to insufficient or poor quality plasma that failed during assessment. Data are presented as mean values ± SEM.

by homology-directed repair and targeted editing in either the tumor suppressor gene *Trp53* (*KrasG12D;Trp53$^{-/-}$*, *KrasG12C; Trp53$^{-/-}$*) or protein kinase *Stk11* (*KrasG12D;Stk11$^{-/-}$*) (Fig. 4a).

As described above, mice were induced with 2 mg/kg dox in the drinking water, and AAV9 carrying sgRNAs were delivered i.v. or by oropharyngeal aspiration (OA) to the lung. All animals that received AAV9 developed adenomas and adenocarcinomas in the lung. OA restricted tumor development to the lungs. Intravenous dosing is a potential method of induction; however, mice dosed i.v. have fewer lung tumors compared to aspiration dosing. Further, i.v. viral dosing also results in unintended tumors in nontarget tissues. We have observed a soft tissue sarcoma present in the nose skin and a cholangiocarcinoma in the liver after i.v. dosing.

A robust tumor development was observed in all models; *KrasG12D;Trp53$^{-/-}$* (Fig. 4b), *KrasG12D;Stk11$^{-/-}$* (Supplementary Fig. 4Aa) and *KrasG12C;Trp53$^{-/-}$* (Supplementary Fig. 4Ba) each with slightly different attributes (Table 2). A key feature of all models is that they are adjustable, the tumor burden can be altered by the viral titer dosed (Supplementary Fig. 4Ae ($n = 30$)).

**ODInCas9 NSCLC models recapitulate clinical tumor pathology.** The tumors generated in the ODInCas9 non-small cell lung cancer (NSCLC) models reflect pathological features of human NSCLC. Multiple tumors were detected in each animal (Fig. 4b and Supplementary Fig. 4A, B). Tumors increase in size over time and as early as 3 weeks post viral infection multiple grade-I and -II bronchial alveolar adenomas were present, progressing to grade-III lung adenocarcinomas and grade-IV adenocarcinomas (Fig. 4b and Supplementary Fig. 4A, B). Commonly, adenocarcinomas are observed infiltrating through the pleura and into the mediastinum. Blood and lymphatic vessel invasion was correlated to distant metastases, primarily to gatekeeper lymph nodes. Tumor and nonneoplastic cells were highly proliferative, by means of Ki67 immunohistochemistry (Fig. 4g and Supplementary Fig. 4Af, 4Be).

Tumor cells expressed pro-surfactant protein C (proSpC) indicative of alveolar type-2 cells as the primary edited cell type (Fig. 4g). All tumors are highly vascularized (Fig. 4 and Supplementary Fig. 4Af, 4Be) and showed a thin aSMA positive stroma. Occasionally, low-grade adenocarcinomas (grade IV) exhibited focal random expression of Tenascin C in tumor cells and extracellular matrix, which was reversely correlated to the loss of Nkx2.1 in these tumor cells indicative of tumor progression (Fig. 4g).

The immunophenotype of the tumor models was characterized by the presence of all relevant major immune cell lines in the tumor or on the surface area in the adjacent lung tissue. Most malignant tumors were macrophage rich with a robust accumulation of F4/80 positive macrophages in the extracellular matrix,

while the benign phenotypes show a varying number of B cells (CD45R) (Fig. 4g and Supplementary Fig. 4Af, 4Be). CD4 T cells and CD8 cytotoxic T cells were observed in low numbers in the tumor stroma (Fig. 4g and Supplementary Figs. 4Af, 4Be).

Eight weeks after viral administration, protein and genomic DNA was extracted from tumor-bearing lungs, and cell lines were derived from tumors isolated from mice transduced with either the *Kras;Trp53$^{-/-}$* and *Kras;Stk11$^{-/-}$* mutations. Western blot analysis for Trp53 expression showed that p53 was not detected in 8/10 cell lines derived from the *Kras;Trp53$^{-/-}$* co-mutated tumors (Fig. 4c). Importantly, *Trp53* cDNA sequencing identified exon 7 deletions in the two Trp53-expressing cell lines (Fig. 4c and data not shown), suggesting that they may express a nonfunctional Trp53 protein. While we have not confirmed that this is non-function Trp53, it demonstrates that in each line *Trp53* was successfully targeted. Trp53 expression was observed in the cell lines derived from tumors where *Kras;Stk11$^{-/-}$* was co-targeted demonstrating selective loss of Trp53 with the appropriate guide RNA. Similarly in the cell lines derived from tumors where *Kras;Stk11$^{-/-}$* were modified, Stk11 protein was not detected, while Stk11 protein was expressed in the cell lines derived from *Kras;Trp53$^{-/-}$* tumors. The cell line data confirm effective and selective targeting of the appropriate proteins. The expression of the target genes was also assessed in whole tumor. Reduction in Trp53 protein was also seen in *Kras;Trp53$^{-/-}$*-targeted tumors relative to normal tissue and *Kras;Stk11$^{-/-}$*-targeted tumors with a 70% and 75% reduction in Trp53 and Stk11 protein relative to normal lung tissue, respectively, confirming the corresponding genotypes in these heterogenous masses of tumor and stromal cells (Fig. 4d). Of note, these results are in line with reports from other NSCLC models[26,27]. Targeted sequencing performed using Illumina MiSeq showed in the *KrasG12D;Trp53$^{-/-}$* model, editing of the *Kras* gene resulting in the oncogenic *Kras* forms G12D, present in 8% of the sequencing reads 28% of the reads had a mutation in the *Trp53* gene (Fig. 4e). In the *KrasG12D;Stk11$^{-/-}$* model, 7 weeks post induction, editing was 7% for *Kras* and 28% for *Stk11* (Supplementary Fig. 4Ac, d). In the *KrasG12C;Trp53$^{-/-}$* model, editing was 13% for *Kras* and 28% for *Trp53* (Supplementary Fig. 4Bd, c). The correct insertion of each specific *Kras* point mutation G12D or G12C was further confirmed by single point mutation sensitive in situ hybridization (BaseScope®, ACD)[28] (Fig. 4f and Supplementary Fig. 4Bd). Further, the tumors, including atypical hyperplasia's express pERK1/2 and pMEK confirming activation of the RAS/MAPK pathway (Fig. 4g and Supplementary Fig. 4Af, 4Be) and correct genome editing.

**Pharmacological interrogation of signaling pathways in ODInCas9 mouse-derived tumor models.** To demonstrate the utility of the ODInCas9 mouse NSCLC models for preclinical testing,

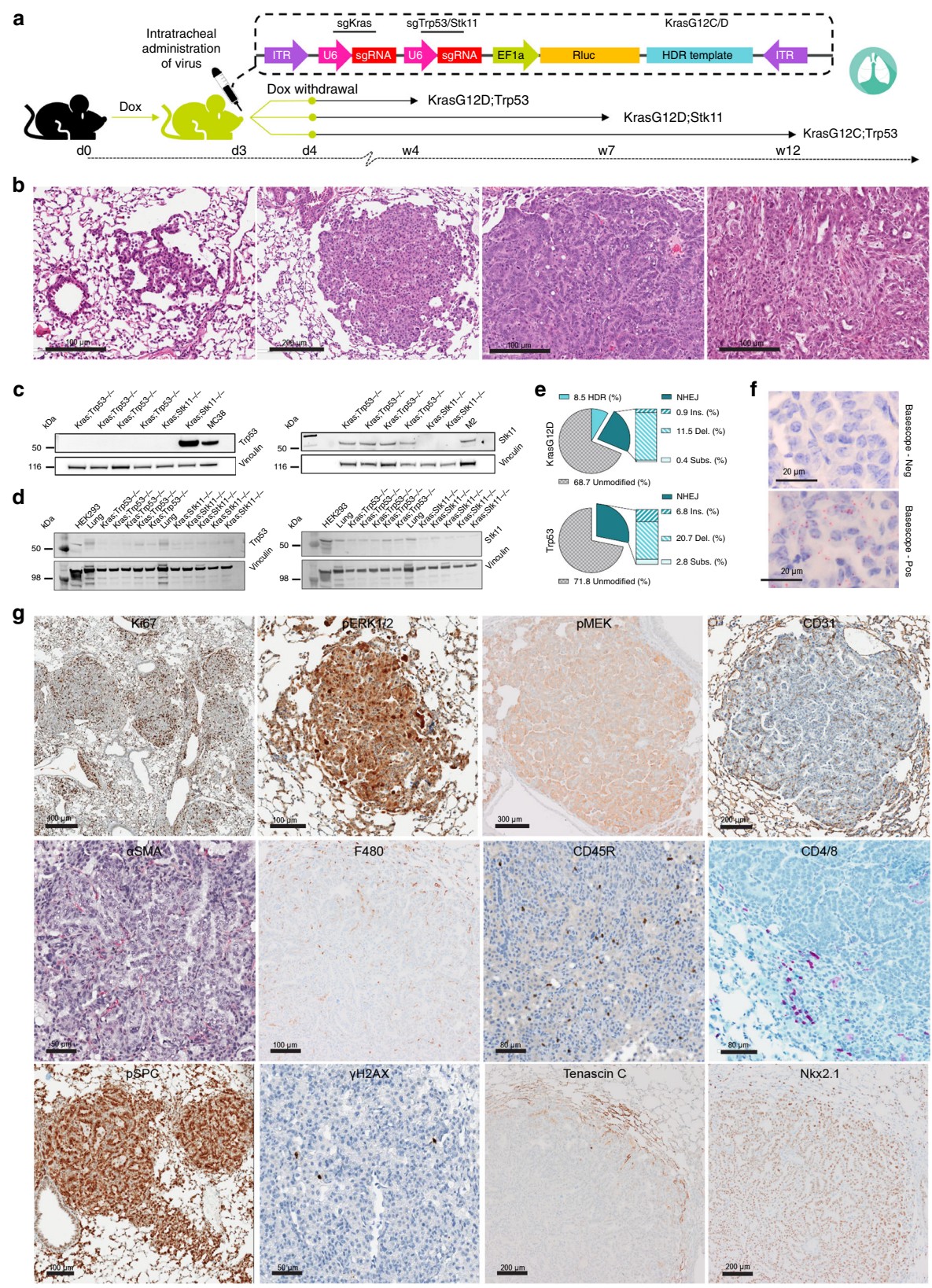

efficacy studies based on previous clinical[29] and preclinical studies[30] were performed (Fig. 5 and Supplementary Fig. 5). ODI-nCas9 mice ($n = 17$) were induced and transduced by AAV9 carrying the sgRNA and homology repair template. *KrasG12D; Trp53*$^{-/-}$ tumors were allowed to progress for 4 weeks prior to treatment with vehicle ($n = 6$) or combination chemotherapy

Docetaxel and the MEK inhibitor AZD6244 (Selumetinib) ($n = 6$) for 2 weeks (Fig. 5a). At 4 weeks there was significant tumor burden as determined by histological assessment of satellite mice sacrificed prior to treatment (Fig. 5a, e). Combination therapy resulted in the stasis of tumor growth, measured by histology at termination, preventing the threefold increase observed in

**Fig. 4 ODInCas9 mouse non-small cell lung cancer models. a** Schematic representation of generation of ODInCas9 KrasG12D;Trp53$^{-/-}$, KrasG12D;Stk11$^{-/-}$, and KrasG12C;Trp53$^{-/-}$ non-small cell lung cancer models. ODInCas9 mice are treated with doxycycline 3 days before administration via oral aspiration of model-specific AAV targeting either Kras by sgRNA and HDR and Trp53 or Stk11 by sgRNA. Doxycycline treatment is kept 1 day post viral administration. Lung adenocarcinoma kinetics is specific for each lung cancer model. **b** Representative HE staining of lung histology capturing the various stages of lung adenocarcinoma. **c** Western blots showing Trp53 and Stk11 expression in cell lines generated from either Kras;Trp53 or Kras;Stk11 lung tumors and **d** in multiple independent lung tumors from the Kras;Trp53 and Kras;Stk11 models. MC38, M2, HEK293 cell lines and non-tumor bearing lung shown as positive control. **e** Classification of amplicon sequencing of lung cancer tissue at Kras and Trp53 locus. **f** DNA in situ hybridization targeting KrasG12D. **g** Immunohistochemistry section of lung tumors staining for Ki67, pERK, pMEK, CD31, αSMA, F480, CD45R, CD4/8, pSPC, γH2AX, Tenascin C, and Nkx2.1.

---

**Table 2 Non-small cell lung cancer (NSCLC) models developed by AAV9 transduction in the ODInCas9 mouse.**

| Genetic alterations | AAV titer | Latency | No. and size of tumor | Features relevant to model |
|---|---|---|---|---|
| KrasG12D;Trp53$^{-/-}$ | $1 \times 10^{11}$ | 4 weeks | Numerous, small | Multiple tumor nodules from 3 weeks post induction, random distribution of nodules of all grades throughout the lobes (Fig. 4b). |
| KrasG12D;Stk11$^{-/-}$ | $1 \times 10^{10}$ | 7 weeks | Few, large | Atypical hyperplasia's, adenomas, and adenocarcinomas at 4 weeks. Fewer tumors per lung lobe with rapid growth of tumors between 12- and 16-week post induction (Supplementary Fig. 4A). |
| KrasG12C;Trp53$^{-/-}$ | $1 \times 10^{11}$ | 12 weeks | Numerous, small | Similar to the KrasG12D;Trp53$^{-/-}$ model; however, these numerous tumor foci progress at a slower rate of growth (Supplementary Fig. 4B). |

---

vehicle-treated animals (one-way ANOVA, $F (2, 14) = 5.449$, $p = 0.0178$) (Fig. 5b, d, e). The majority of tumors responded to combination therapy, particularly the inhibition of the RAS/MAPK pathway via inhibition of MEK1/2 as demonstrated by decreased pERK staining. Further, measurement of tumor volume by MRI confirmed stasis of tumor progression with a significant treatment effect (two-way ANOVA, $F (1, 19) = 5.475$, $p = 0.0304$), one vehicle mouse was euthanased prior to the second MRI scan (Fig. 5c, f).

ODInCas9 mice ($n = 14$) were dox-induced and -transduced by AAV9. Twelve weeks after KrasG12C;Trp53$^{-/-}$ induction mice were randomized to treatment arm and treated with either vehicle ($n = 4$) or combination chemotherapy Docetaxel and MEK inhibitor AZD6244 (Selumetinib) ($n = 5$) for 4 weeks. Combination treatment resulted in stasis of tumor progression and significantly decreased the percentage of lung occupied by tumor (one-way ANOVA, $F (2, 11) = 5.756$, $p = 0.0195$) (Supplementary Fig. 5b, d).

These results demonstrate that the ODInCas9 mouse can be used to generate genetically complex models. In this instance we have with CRISPR-Cas9 editing created both an activating mutation in Kras and loss of function in Trp53. Further, these models have tumors that are clinically relevant, developed in a timescale conducive to therapeutic intervention, and they are susceptible to pathway blockade and chemotherapy.

## Discussion

When using Cas9-mediated gene editing to generate genetically engineered models, constitutive or leaky expression of Cas9 can have a negative impact on the utility of the model. Since short-induced expression of Cas9 is important to limit off-target gene editing[31,32], our aim was to create a sensitive but tightly regulated all-in-one inducible Cas9 system that could be used in a variety of genomic backgrounds of human and mouse origins as well as generating an inducible mouse.

As the ODInCas9 system is an all-in-one system it allows for rapid cell line generation by a single transgene integration event compared to split Tet-On element systems[33]. In contrast to some previous all-in-one system designs[33] the ODInCas9 system shows tight and strong activation due to the insulator design allowing

for the use of a strong CAG promoter without leakage. Moreover, in comparison to other inducible systems[34], the use of ObLiGaRe allowed for targeted integration of the large transgene (ODInCas9 cassette) by ZFN homologies in both human and mouse genomic backgrounds, in parallel.

Our results indicate that the ODInCas9 system is tightly regulated with no leakage detected in the cell lines or, in the mouse. In all methodologies used to investigate Cas9 expression without dox induction, non-induced cells, and mouse tissue (not exposed to dox) was always at the lowest detectable limits of the experimental system used. The ODInCas9 system is very sensitive (1–5-ng/μl dox activation) and tunable. Moreover, as the system turns off upon withdrawal of dox it can be repeatedly activated for sequential editing of multiple genes, but more importantly inactivation minimizes unwanted downstream effects of the system and Cas9 exposure enabling purer efficacy studies. This emphasizes the advantage of the ODInCas9-inducible model over Cre-activated Cas9-inducible systems that are constitutively active upon recombination[18,19,35,36]. Tight regulation of Cas9 expression in the ODInCas9 system overcomes the leaky nature of Cre or Cas9 reported in other models[18]. Further, the Cre-loxP system is limited by the presence of cryptic loxP sites in the mammalian genome, thus genotoxicity is induced by Cre-recombinase[22,37], which is not observed with the dox Tet-inducible systems.

As the ODInCas9 transgenic mouse shows tight control of Cas9 expression, it offers an approach to generate tumor models through direct modification of multiple oncogenes. Focusing on NSCLC, it has been possible to rapidly generate three genetically distinct tumor models, by specific modification of KrasG12D and KrasG12C, in conjunction with either Trp53 or Stk11. Each of the models generated tumors with a latency of 4–12 weeks and within the course of tumor experiments, health status of the animal was only impacted by disease burden and not by nonspecific side effects of the Cas9 system. As expected we observed tumors in nontarget tissues by using i.v. injection of virus but tissue-specific delivery like OA restricted tumor development to the lungs. Therefore tissue-specific delivery is one way to achieve spatial control. An alternative way would be to express the rtTA under a tissue-specific promoter as described by Schönig and Bujard[38].

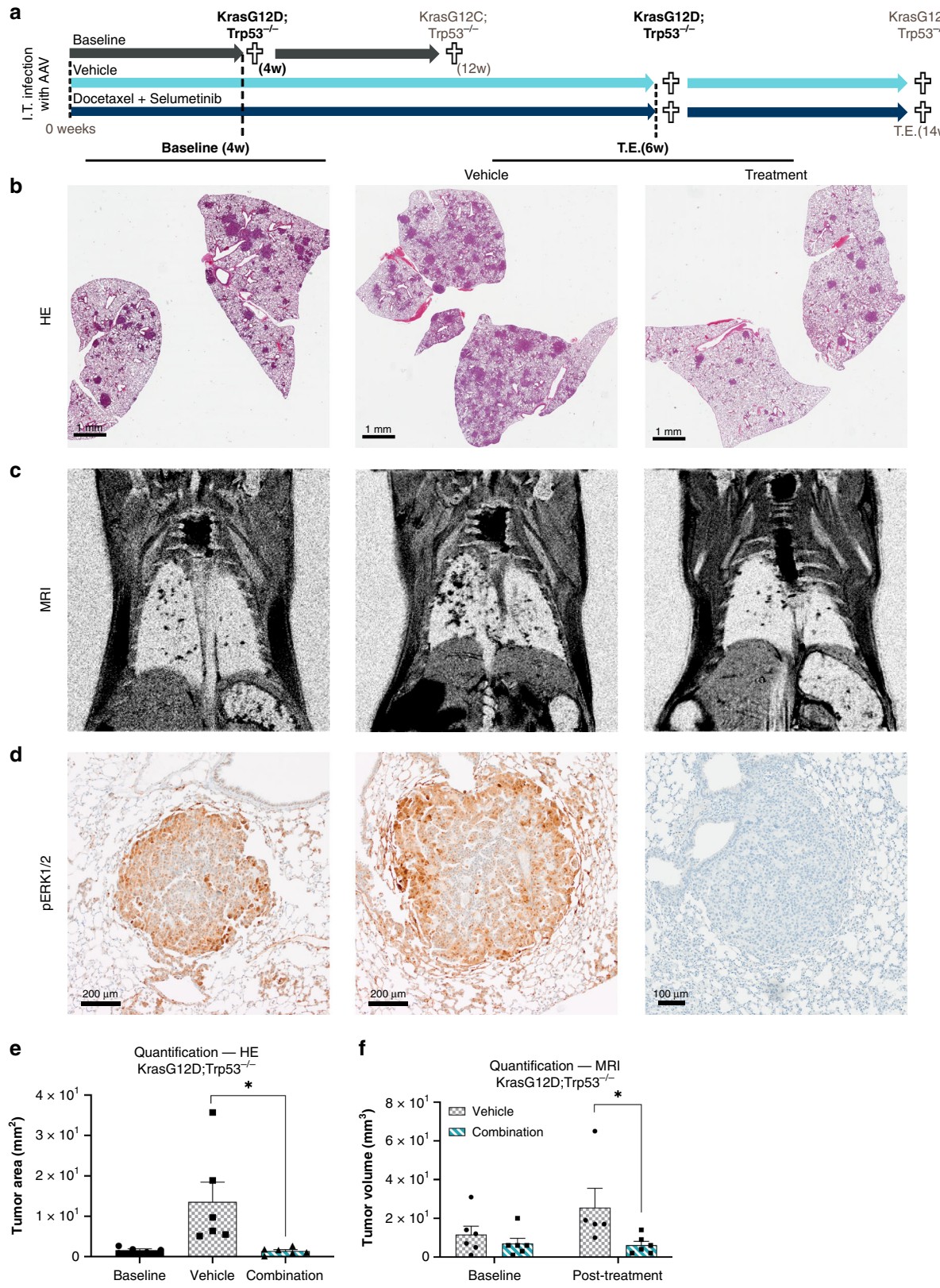

Finally it is possible to achieve spatial control of CRISPR by combining miR expression with anti-CRISPR regulation as described by Wang et al.[39].

We demonstrate the successful generation of three models of non-small cell lung cancer. Histological and immunohistological assessment demonstrated development of multiple tumor nodes in the lungs. Tumor formation in combination with efficacy studies, including treatment with pathway-specific inhibitors, significantly decreasing tumor burden suggests that the ODI-nCas9 mouse is a suitable replacement for traditional GEMMs.

**Fig. 5 Pre-clinical efficacy study in ODInCas9 mouse non-small cell lung cancer model. a** Timelines for treatment studies. ODInCas9 mice were induced with dox, dosed with AAV, and allocated to treatment arm at 4 weeks KrasG12D;Trp53$^{-/-}$ and 12 weeks KrasG12C;Trp53$^{-/-}$, when significant tumor burden is observed. KrasG12D;Trp53$^{-/-}$ mice were either killed at 4 weeks for baseline tumor burden or treated for 4 weeks with either vehicle or Docetaxel + Selumetinib combination treatment. Evaluation at baseline and treatment end (T.E) by **b** lung tissue histology (HE), **c** MRI, **d** pERK1/2 immunohistochemical staining. Quantification of tumor burden at baseline, vehicle, and combinatory treatment using **e** HE histology (one-way ANOVA, $F_{(2, 14)} = 5.449$, $p = 0.0178$) and **f** MRI (two-way ANOVA, $F_{(1, 19)} = 5.475$, $p = 0.0304$). Experimental set-up included $n = 6$ animals per group except HE baseline tumor burden $n = 5$. MRI vehicle post-treatment had $n = 5$ as mouse was euthanased due to ethical guidelines. Data are presented as mean values ± SEM.

Further, transgenic mice bearing the ODInCas9 transgene can be used to generate genotype-specific models at scale and in a coordinated time frame, enabling time-controlled experiments to be performed with many animals in an individual induction cohort. Ideally these complex and long-term in vivo experiments should edit the genome within a specific window to govern the time frame of induction of the genomic manipulations.

Unlike other tumor GEMMs, the ODInCas9 model enables development of multiple tumor models, with genetically modified alleles, from a single mouse strain. As such, this allows for a significant reduction in the time and number of animals required for the generation of each tumor model. Further, NSCLC tumor development in the ODInCas9 mouse is faster than traditional GEMMs. In some ODInCas9 models, tumors, including adenocarcinomas, can be found ~4 weeks after induction. This allows for treatment studies to be initiated at that time, rather than months after induction, as with previous lung GEMMs that targeted the same loci[7,40–42]. Importantly, tumor burden can be easily controlled to match ideal treatment schedules through regulation of the number of infective viral titer administered.

Analysis of individual lesions revealed heterogeneous mutation patterns within the *Kras* allele. There was between 8 and 14% correct incorporation of the homology arm into the cut site as measured by next-generation sequencing. The mRNA for each of the point mutations (*Kras*G12D and *Kras*G12C) was expressed in ~50% of cells within an adenocarcinoma carrying at least one copy of the gene. The difference in the indolence of each of the models, *Kras*G12D vs. G12C and the *Trp53* with *Stk11* reveals a heterogeneity that has been reported by others[13,18,21] and likely results from differential usage of CRISPR-Cas9 double-strand break repair mechanisms, homology-directed repair, or non-homologous end joining, followed by different selective pressures over the heterogenous population of genetic mutations that arise in each model. This heterogeneity is unlike the results reported for other GEMMs and is an advantage of the inducible Cas9 systems[43]. In contrast to previous reports[44], where disruption of *Trp53* prevents the DNA damage response and increases the rate of homologous recombination, the ODInCas9 system had a similar degree of genome modification by homology-directed repair to *Kras* with either *Stk11* or *Trp53* editing.

In summary, the ODInCas9 system offers a fast, reproducible, and tunable genetic engineering development platform, containing a tightly controlled inducible Cas9. This model can support the preclinical pipeline with fast cell line generation (2 weeks) and in vivo efficacy end points as one mouse enables the development of multiple models. Further, the ability to perform multiple rounds of modification and the potential for in vitro, ex vivo, and in vivo modification add to the flexibility of the ODInCas9 system enabling quick KO generation for target validation and CRISPR screening. The ability to precisely edit at different time intervals allows for preclinical replication of complex tumor mutational burden observed in oncology. Moreover, this system will decrease cost, time and reduce animal numbers compared to more traditional GEMMs, as genetic manipulation of multiple loci can be achieved from a single delivery of sgRNA and homology template.

## Methods

**Plasmids.** Sequences of both the Dual AAVS1 ZFN and the ODInCas9 construct (containing SpCas9) are provided in Supplementary Notes 1 and 2, respectively.

**Cell culture.** HEK293, HCT116, HepG2, OVACAR8, A549, and Neuro2A were all purchased from ATCC (Manassas, VA, USA) and maintained in DMEM high glucose supplemented with 10% FBS, NEAA (1:100), and 1% penicillin–streptomycin (all from Invitrogen, Carlsbad, CA). HCT116, OVACAR8, A549, and Neuro2A were grown on tissue-culture-treated plates and HEK293 and HepG2, on poly-L-ornithine (20 μg/cm², Sigma, St Louise, MO) coated plates. hiPSC were generated and maintained in a feeder-free human pluripotency culturing system, Cellartis DEF-CS 500 (Takara, Japan), according to manufacturer's instructions[45]. M2-10B4 (M2) and MC38 were used as positive controls for Western blots. M2 is a clone derived from murine bone marrow stromal cells with a fibroblast morphology. MC38 is a murine colon adenocarcinoma cell line generated from C57BL/6 mice that highly express mouse p53 protein. All cell lines were cultured at 37 °C and 5% $CO_2$ and routinely passage when reaching 80% confluency.

**ODInCas9 cell line generation.** *Transfection*: For ODInCas9 cell line generation of HEK293, HCT116, HepG2, Neuro2A, A549, OVCAR8, and hiPSC, transfection was performed at 80% confluency. Total DNA of 500 ng of ODInCas9 plasmid together with ZFN-AAVS1 plasmid at a ratio of 1:10 was mixed and transfected by lipid-based transfection using FuGene HD (Promega, Madison, WI) according to manufacturer's instructions.

*Transgene integration selection*: For transgene integration selection, HEK293, HCT116, HepG2, Neuro2A transfected cells were selected with G418 (Sigma) at a concentration of 100 μg/ml starting 5 days after transfection for the following 7–9 days before single cell sorted using BD FACSAria II (BD Bioscience, San Jose, CA) for clonal cell line generation. For transgene integration selection of A549 and OVCAR8, cells were Dox (100 ng/ml) treated 24 h before FACS purification of GFP expressing cells for the generation of ODInCas9 pools. A549 clonal line was generated by subsequent single-cell dilution (0.75 cells/w) into 384wp after FACS purification. For transgene integration selection of hiPSC, the cells were selected with G418 (Sigma) at a concentration of 50 μg/ml starting 5 days after transfection for the following 9 days. Human iPSC clonal generation was performed by single-cell dilution (0.75 cells/w) into 384wp.

*Cell line validation*: ODInCas9 HEK293, HCT116, HepG2, Neuro2A, A549, and OVCAR8 cell line validation was performed by 24–48 h Dox induction at 10 μg/ml following GFP detection by microscopic measurement using ImageXpress Micro XLS Widefield Microscope (Molecular Devices, Sunnyvale, CA) or Incucyte Zoom (Essen Bioscience, Michigan, US). Validation of transgene editing function across the ODInCas9 cells lines HEK293-C62, HCT116-12.1, HepG2-Pool, Neuro2A-C16, A549-C10, OVCAR8-Pool, and hiPSC-C86 was performed by lipid-based transfection (FuGene HD) of sgRNAs expressing plasmids targeting MCT1 for OVCAR8, Trp53 for N2A, and GFAP for HEK293, HCT116, HepG2, and A549. Cells were Dox (10 μg/ml) treated 12–24 h before transfection followed by continuous Dox treatment until end point 48 h post transfection.

For generation of nonactivated ODInCas9 hiPSC cell lines, confluent clonal 384 wells were split into three subsequent plates for validation of transgene activation, transgene AAVS1 locus integration, transgene copy number integration, and continues subculturing. Validation of transgene activation cells was treated with Dox (10 μg/ml) for 24–48 h following GFP detection using ImageXpress Micro XLS Widefield Microscope (Molecular Devices, Sunnyvale, CA). Validation of transgene locus insertion was validated by junction and AAVS1 locus PCR.

Transgene copy number was evaluated by a droplet digital PCR assay (ddPCR) purchased from BioRAD targeting the SpCas9 sequence of the ODInCas9 plasmid and normalized to a reference probe AP3B1. Digital-droplet PCR was performed according to manufacturer's instructions (Bio-Rad, Hercules, CA). Lipid droplets are generated in part from a 20 μl PCR reaction containing 100 ng genomic DNA including Cas9 primers and probes. The lipid droplets undergo a cycling PCR program before being analyzed using a droplet reader (QX200, BioRAD). The number of positive droplets generated by the Cas9 ddPCR assay is normalized to a reference probe AP3B1 to assess transgene copy number.

Quality of the ODInCas9 hiPSC clones was assessed by karyotyping, pluripotency marker expression, and differentiation capacity. Karyotyping was evaluated by g-banding using Cell Guidance Systems karyotyping analysis service (Cell Guidance Systems, Cambridge, UK). Cell fixation was done according to instruction by Cell Guidance Systems and then shipped for analysis.

Analysis of pluripotency marker expression of hiPSC ODInCas9 clones was performed using the BD Stemflow kit (BD Bioscience). Cells were detached using TryPLE and resuspended in before PBS before cell fixation using Cytofix (BD). Cells were permeabilized and stained with Alexa Fluor® 647 Mouse anti-SSEA-4 (clone: MC813) and PerCP-Cy™5.5 Mouse anti-Oct-3/4 (clone: Clone: 40/Oct-3) including isotype controls Alexa Fluor® 647 Mouse IgG3, κ Isotype Control (Clone: J606) and PerCP-Cy5.5 Mouse IgG1, κ Isotype Control (Clone: X40). Stained cells were resuspended in FACS buffer (PBS + 2% FBS) and analyzed by FACS using BD LSRFortessa (BD Bioscience).

Differentiation capacity was assessed by applying the STEMdiff™ Trilineage Differentiation Kit. iPSCs were adapted from DEF-CS culturing system to mTeSR™1 (STEMcell Technologies) culturing system by seeding 40,000/cm$^2$ on Geltrex (ThermoFisher) for 4 days before trilineage differentiation start. Cells were seeded into three different conditions: ectoderm, mesoderm, and endoderm at 200,000/cm$^2$, 50,000/cm$^2$, and 200,000 cells/cm$^2$, respectively. One day post passage cultures where switch to respective differentiation media and cultured 5 days for mesoderm and endoderm lineage differentiation or 7 days for ectoderm lineage differentiation. Medium was changed daily. Cells were fixed and stained for SOX1 (AF3369, R&D Systems), SOX17 (562205, BD Bioscience), and Brachyury (X1AO2, eBioscience) assessing ectoderm, endoderm, and mesoderm differentiation capacity, respectively.

*ODInCas9 transgene induction evaluation*: Induction titration studies of the ODInCas9 cell lines; HEK293-C12.1, A549-C10, HepG2-Pool, hiPSC C25/C86 ranged from 1 pg/ml to 100 ng/ml and were applied for 24 h before end point assessment at 48 h looking at either GFP expression and/or Cas9 protein concentrations. Activation time course study of the hiPSC ODInCas9 C86 clonal line applied 50 ng/ml continuously for 48 h during live cell imaging. Inactivation time course study of the ODInCas9 construct in hiPSC ODInCas9 C86 clonal line was performed by a pulse induction of 10 μg/ml for 1 h followed by 3× PBS wash where cells were imaged and collected every day for GFP detection and Cas9 protein for 7 days.

Evaluation of cell population activation of ODInCas9 lines hiPSC-86, HEK293-C12.1, and N2A-C16 was performed 48 h post Dox induction (0, 0.1, 1, and 10 μg/ml). Cells were washed with PBS, detached in 2-mM EDTA (in 1× PBS) and resuspended in FACS buffer (2-mM EDTA, 2% FBS, 1× PBS). Cells were transferred in 96-well plates (Greiner #651201) and subjected to flow cytometry on an iQue Screener PLUS (Intellicyt). GFP positive cells were identified using a BL1 detector (530 nm) and data were analyzed with the ForeCyt software.

Evaluation of transgene transcriptional activation in ODInCas9 C25/C86 lines was performed by collecting RNA 48 h post experimental start of dox-treated (10 μg/ml) and non-treated cells. Total RNA was isolated using the miRNeasy Kit (Qiagen). The quality of the RNA was assessed by a Fragment Analyzer (Advanced Analytical Technologies, Ankeny, IA). Samples with RNA integrity number >9 were used for library preparation. One microgram of total RNA was used for long RNA library construction. Illumina TrueSeq Stranded mRNA LT Sample Prep Kit (Illumina, San Diego, CA) was used to construct poly(A)-selected paired-end sequencing libraries according to TrueSeq Stranded mRNA Sample Preparation Guide (Illumina). All libraries were quantified with the Fragment Analyzer (Advanced Analytical Technologies), pooled and quantified with Qubit Fluorometer (Invitrogen), and sequenced using Illumina NextSeq 500 sequencer (Illumina). Three biological replicates were sequenced per sample. For RNAseq data analysis Salmon was used to quantify transcript read counts[46]. A hybrid reference transcript was generated using Ensemble v94 and Tet-On 3G and Cas9 sequences located in the plasmid. Transcript level data were collapsed to the gene level and then normalized using tximport[47].

*Evaluation of sgRNA delivery*: Single gRNAs were transfected either as plasmid, in vitro transcribed or synthetic. Induction with Dox (100 ng/ml) was performed 12–24 h before transfection and maintained to end of experiment, 48 h post transfection. Cells were grown to 80% confluency and while passaged mixed with transfection reagents before seeding. End point experimental readout was to evaluate indel (see indel assessment method section) and/or protein expression (see western blot method section). Plasmid transfection was performed using FuGene, transfecting 500 ng plasmid per 24 w having a reagent:DNA ration of 3.5:1. DNA was added to 26 μl DMEM Optimem (Invitrogen) and mixed with 1.9 μl FuGene reagent. Complex formation was incubated 10–15 min in RT before mixed with 40,000 ODInCas9 C86 cells seeded into a 24 w. Plasmid sgRNAs targeted GFAP (GFAP-Cr1,2).

Synthetic two component sgRNAs in the form of crRNA and tracrRNA were purchased from IDT (Integrated DNA Technologies, Coralville, IA) (Supplementary Table 2). Preparation of synthetic sgRNAs by preparing 100 μM stock concentrations of crRNA and tracrRNA, respectively, before mixing crRNA, tracrRNA, and Duplex Buffer (1:1:10). RNA mixture was heated for 5 min at 95° C followed by cooling to room temperature. RNA was delivered using either lipid-based transfection (RNAiMAX) or electroporation. RNAiMAX (Invitrogen) and RNA was first mixed with DMEM Optimem (Invitrogen), separately, before combining the two solutions followed by a 20 min incubation. ODInCas9 C86 cells were mixed with the lipid complex solution before seeding. For single sgRNA targeting (PIGM-Cr1,2) 30 pmol RNA mixture was transfected per 40,000 cells while for dual sgRNA targeting (PIGM-Cr1&2, GFAP-Cr1&2) 15 pmol RNA mixture per target was transfected per 40,000 cells. RNA delivery using the NEON electroporation system (ThermoFisher) was performed by applying 1600 v, 10 ms and 3 pulses of OVARC8 ODInCas9 pool mixed with RNA, 50,000 cells and 1 μL of sgRNA (MCT-Cr1& 2) per 10 μL tip.

In vitro transcribed sgRNA (GFAP-Cr1&2) was purchased from Eupheria Biotech (Dresden, Germany) and loaded into LNPs. Different amounts of LNPs (100–4000 ng) were mixed with 40,000 ODInCas9 C86 cells and seeded into a 24w.

*Proliferation imaging*: Human iPSC ODInCas9 line C86 was either non-induced or pulse induced (1 h with 10 μg/ml Dox following washout) 6 h before synthetic sgRNA transfection using GFAP and Alu targets (Alu-Cr1,2, GFAP-Cr1&2). Growth curves were measured using IncuCyte Zoom microscope (Essen Bioscience Inc., Ann Arbor, MI) for 42 h post transfection. OVCAR8 ODInCas9 cells were induced 24 h before transfection and kept at 100 ng/mL Dox during the experiment. Transfection of synthetic spCas9 sgRNAs targeting MCT1 (MCT1-Cr1&2) or CDK12 (CDK12-Cr1&2) in dual combination was performed. Alu targets (Alu-cr1,2) were used as viability control. All targets were performed in triplicates.

*FLAER assay*: FLAER assay (Cedarlane, Canada) tags mammalian GPI anchors through an AlexaFluor488-labeled version of aerolysin. Human iPSC ODInCas9 line C86 was pulse induced 1 h with 10 μg/ml Dox followed by washout 6 h before standard synthetic sgRNA transfection targeting three sites of the *PIGM* gene (PIGM-Cr1,2,3) in single and dual combination. ODInCas9 system was left to turn off for 7 days post transfection before assessment using FACS instrument BD LSRFortessa (BD Bioscience) and confocal microscopy instrument Cell Voyager 7000S (CV7000S, Yokogawa, Japan). Cells were detached using TryPLE (Invitrogen) and washed before cell suspension was incubated at room temperature for 15 min at a final FLAER concentration of 0.5 μM in 100 μl volume. Cells were washed twice and resuspended in 100 μl DPBS containing 2% fetal bovine serum (both from Invitrogen) before 5 μl was used to plate cells for microscopy imaging and the rest used for FACS analysis. Data were subsequently analyzed using FlowJo to assess reduction of GFP expression.

*On off activation of ODInCas9 transgene*: Validation of on off capacity of the ODInCas9 transgene hiPSC ODInCas9 C86 was either non-induced or pulse induced 1 h with 10 μg/ml Dox followed by washout 6 h before transfected with dual synthetic sgRNA transfection targeting *GFAP* (GFAP-Cr1&2) at d0. Induced cells at d0 were divided at d4 into non-induced or 2nd pulse induced (1 h with 10 μg/ml Dox followed by washout) cells before transfected with dual synthetic sgRNA transfection targeting *MYLIP* (MYLIP-Cr1&2). Non-induced cells at d0 were also transfected with dual synthetic sgRNA transfection targeting *MYLIP* and cultured to d7. DNA samples were collected at d4 and d7 for indel assessment (see Indel/edit detection and quantification by nuclease assay and next-generation sequencing). Protein collection was performed daily from d1 to d8 to evaluate Cas9 protein expression (see SDS-PAGE and Western blot analysis).

*Whole transcriptome profiling by RNA sequencing and bioinformatics*: Total RNA of 2–3 million cells was isolated using the miRNeasy Kit (Qiagen). The quality of the RNA was assessed by a standard sensitivity NGS fragment analysis kit on Fragment Analyzer (Advanced Analytical Technologies). All the samples had RNA integrity number >9.8 and were used for library preparation. One microgram of total RNA was used for each library. Illumina TruSeq Stranded Total RNA Ribo-Zero Human/Mouse/Rat Gold (Illumina) was used to construct ribosomal RNA depleted sequencing libraries. All libraries were quantified with the Fragment Analyzer, pooled in equimolar concentrations, and quantified with Qubit Fluorometer (Invitrogen). Libraries sequenced >40 M paired-end reads using the High Output Kit v2 (150 cycles) on an Illumina NextSeq 500. Three biological replicates were sequenced per sample.

RNA-seq fastq files were processed using bcbio-nextgen (version 0.9.7) where reads were mapped to the genome with between 23.9 and 119.3 M mapped reads per sample (with a mean of 69.3 M). Gene level quantifications, counts, and transcript per million (TPM), were generated with featurecounts (version 1.4.4) and sailfish (version 0.9.0), respectively, all within bcbio. All analyses were performed using R (version 3.4.0, https://www.r-project.org/). Differential gene expression were assessed with DESeq2 (version 1.14.1).

*Indel/edit detection and quantification by nuclease assay and next-generation sequencing*: Gene editing was assessed by mismatch-specific endonuclease activity, either T7 endonuclease I (T7EI) or surveyor nuclease, on PCR amplified sequence of the target locus (Supplementary Table 3). Genomic DNA was isolated using QuickExtract DNA Extraction Solution (Lucigen). PCR amplified target sequence was heated to 95 °C and slowly (2 °C/s) cooled down to 25 °C to form heteroduplexes before incubation with mismatch-specific endonuclease, which cleaves heteroduplexes enabling detection of indel formation. Cleaved products were analyzed by DNA electrophoresis using Novex TBE Gel 10% (ThermoFisher) or QIAxcel (Qiagen). Quantification of detected gene edit is performed as previously described[48].

Quantification of genetic modifications was performed by next-generation sequencing method amplicon sequencing. Amplicons were generated by amplifying the genomic target region by using PCR primers linked to sequences of Illumina Nextera Adapters. A primary adapter 15 μl PCR reaction was carried out using 2 μl of genomic material from cell lysate of a confluent 96 well together FusionFlash High Fidelity Master Mix (Thermo). PCR reaction was cleaned up with Agencourt AmPure XP beads (Beckman Coulter). A secondary indexing PCR was performed using Illumina Indexes followed by a bead purification as described previously. The libraries were quantified using Fragment Analyzer (Advanced Analytical Technologies). The indexed libraries were pooled and subjected to paired-end sequencing using Illumina NextSeq 500 mid-output with a 150 bp read length. Sequencing output was analyzed using RIMA[49]. Briefly, the reads were merged and mapped to the amplicon reference sequence. Variants (excluding the single

nucleotide variations) were called with a minimum allele frequency of 0.1% and a minimum base quality of 25 Phred 33 scores. The resulting variants were analyzed using RIMA. Variants not overlapping of the cutting window (±2 base pairs of the cut site) were excluded. Finally, the editing efficiencies were measured as the percentage of modified reads in mapped reads.

*Immunocytochemistry*: Twenty-four hours after 100 ng/mL dox treatment, the OVCAR8 ODInCas9 cells were fixed with methanol followed by permeabilization with 0.1% Triton X100. The samples were blocked in 10% goat serum and incubated overnight at 4 °C with the mouse Cas9 antibody (7A9-3A3 Cell Signaling Technologies) 1:800 dilution in 1% BSA (Supplementary Table 4). After washing, the cells were incubated with the goat anti-mouse DyLightTM 488 secondary antibody (ThermoFisher) 1:200 in 1% BSA. DAPI (0.5 μg/ml) was used to stain the nucleus of the cells. The stained cells were analyzed by the ImageXpress confocal microscope.

*Generation of the OdinCas9 mouse*: The target construct was electroporated into C57Bl/6N (Prx) ES cells. Neo-resistant clones were analyzed for correct integration into the R26 locus by PCR. Validation of transgene induction in mESC ODInCas9 cells was done by 48 h Dox (10 μg/ml) treatment assessed by GFP fluorescent signal and Cas9 protein expression. Three positive clones were further screened by Southern blot, using a Neo probe. One validated clone was expanded and injected into Balb/cAnNCrl blastocysts to generate chimeric mice. Chimeric C57Bl/6N OdinCas9 heterozygous males were bred to C57Bl/6NCrl females to generate experimental animals. Litters are genotyped using the following primers; ACGTTTCCGACTTGAGTTGC and GTGCAATCCATCTTGTTCA.

*Animals*: All mouse experiments were approved by the AstraZeneca internal committee for animal studies and the Gothenburg Ethics Committee for Experimental Animals (license numbers: 162-2015+ and 629-2017) compliant with EU directives on the protection of animals used for scientific purposes. Experimental heterozygous mice were generated by breeding male heterozygous ODinCas9 mice to female C57Bl/6NCrl mice (Charles River). Male and female ODinCas9 were bred to generate homozygous mice to understand the inheritance pattern of the cassette and to generate an experimental cohort. OdinCas9 mice were crossed to the described human knock in PCSK9 mice[25,50] to generate mice heterozygous for both constructs. In all experiments mice were randomized to groups to ensure each group had similar body weight. Mice were housed in a temperature controlled room (21 °C) with a 12:12 h light–dark cycle (dawn: 5.30 a. m., lights on: 6.00 a.m., dusk: 5.30 p.m., lights off: 6 p.m.) and with controlled humidity (45–55%). Mice had access to a normal chow diet (R70, Lactamin AB, Stockholm, Sweden) and water ad libitum. Housing included environmental enrichment such as cardboard tubes, shredded paper, and chew sticks. Mice were checked daily and weighed weekly.

*Guide RNA design and adeno-associated viral constructs*: AAV (serotype 9) were custom generated by Vector Biolabs (Malvern, PA, USA). Each AAV consisted of two sgRNAs under a U6 promotor and contained a point mutation-specific homology arm for each KrasG12D or G12C. The guide RNAs used for the NSCLC studies have been validated previously (Supplementary Table 4)[13].

*CRISPR-Cas9 induced editing with LNP or AAV in ODinCas9 mice*: Induction of Cas9 required dox hyclate (2 mg/ml; Sigma Aldrich, MI, USA) supplemented with 1% sucrose added to the drinking water for three nights. No significant weight loss, hepatocellular necrosis, or changes in liver or GI tract morphology were observed with drinking water administration of dox. To test the efficacy of guide delivery to the liver, OdinCas9 mice crossed with the human PCSK9 mice were injected i.v. with LNP containing guide for PCSK9[25,50]. To generate models of NSCLC ODInCas9 mice (12–16 weeks), after two nights on Dox, mice were dosed by OA with $1 \times 10^{10}$ GC (*KrasG12D;Stk11⁻/⁻*), or $1 \times 10^{11}$ GC (*KrasG12D;Trp53⁻/⁻, KrasG12C;Trp53⁻/⁻*) genome copies of adeno-associated (AAV9) diluted to a final volume of 50 μl in PBS. Twenty-four hours post dosing the mice were returned to normal drinking water.

*NSCLC treatment studies*: KrasG12D;Trp53⁻/⁻ ODInCas9 mice (16 weeks, male $n = 18$, $33.5 \pm 2$ g) had two nights of dox, then dosed by OA with $1 \times 10^{11}$ GC of AAV9. Mice were weighed weekly and randomized to treatment group after 4 weeks ensuring each group had matched body weight. For 2 weeks, mice received Selumetinib (AZD6244, 25 mg/kg, resuspended in 0.5% HPMC/0.1% Tween 80) by oral gavage (BID, 8 h apart). Once a week, 1 h after Selumetinib dosing, Docetaxel (Taxotere, 15 mg/kg, in physiological saline) was administered i.v., *KrasG12C; Trp53⁻/⁻* ODInCas9 mice (10–12 weeks, male $n = 9$, $28.7 \pm 2$ g and female $n = 13$, $22.6 \pm 1$ g) had two nights dox and were OA dosed $1 \times 10^{11}$ GC of AAV9. Mice were weighed weekly and randomized (three male and five female) per treatment group after 12 weeks ensuring each treatment arm had matching body weight. For 4 weeks, mice PO dosed (BID, 8 h apart) with Selumetinib (AZD6244, 25 mg/ kg). Once a week, 1 h after Selumetinib, Docetaxel (Taxotere, 15 mg/kg) was administered i.v.

*Histology and immunohistochemistry*: At necropsy, mice were euthanized under isoflurane anesthesia, and lung, pancreas, liver, skeletal muscle, kidney, brain, gastrointestinal tract, reproductive system and spleen were collected in 4% neutral buffered formalin for assessment of Cas9 immunostained cell distribution. Further, lung tissue was inflated with 4% neutral buffered formalin and with other tissues immediately fixed in 4% neutral buffered formalin. Tissue was embedded in paraffin and prepared as 5 μm thick sections. Sections were stained for hematoxylin and eosin for morphological characterization. To define the extracellular matrix, immune phenotype, pathway activation, and the proliferative nature of the tumors, immunohistochemistry was performed with antibodies for αSMA, tenascin C,

Nkx2.1, proSpC, CD31, F480, CD45R, CD4/8, pERK1/2, pMEK, Ki67, γH2AX (supplier details and RRID# in Supplementary Table 4). Immunohistochemistry was performed using the Discovery Ultra (Ventana Medical Systems, Inc, AZ, USA). Antigen retrieval using cell condition fluid 1 was performed when required and the Discovery ChromoMap DAB was used as the detection system for all stains except CD4 Discovery Purple HRP and CD8 Discovery Teal HRP. All histological slides were blinded and examined using light microscopy (Carl Zeiss Microscopy GmbH, Jena, Germany) by an experienced board-certified pathologist. The tumor classification occurred as previously described and included atypical hyperplasia (grade 1), adenoma (grade 2), high-grade adenocarcinoma (grade 3), and low-grade adenocarcinoma (grade 4)[12].

*Indel/edit quantification by next-generation sequencing*: Genomic DNA was extracted from whole lung tissue and isolated single adenocarcinomas from AAV injected mice using the Qiagen Puregene cell and tissue kit (Qiagen, Germany). The genomic DNA was amplified with adapter-containing gene-specific primers (refer to Supplementary Table 3), linker for forward primers: TCGTCGCAGC GTCAGATGTGTATAAGAGACAG; the linker for reverse primers: GTCTCGTG GGCTCGGAGATGTGTATAAGAGACAG using Q5 Hot Start High Fidelity DNA polymerase (NEB) and sequenced by using a NextSeq 500 Instrument (Illumina, California, United States). Read counts above 10 K were achieved for the in vivo Trp53 and LNP delivery of guides. Read counts of 50 K were achieved for all other analyzed alleles. Sequencing reads were demultiplexed using Illumina software and r1 and r2 FASTQ files were analyzed using CRISPRResso[51]. Briefly, reads with a minimum average quality score of 30 were aligned to the reference sequence. A window of 30 base pairs centered on the predicted cleavage site was specified for the quantification of indels and base-editing outcome.

The frequencies of indel mutations were automatically calculated using CRISPResso. The frequency of mutated alleles was calculated based on the CRISPResso's detected alleles, "Alleles_frequency_table.txt," in NGS data as follows: first, all the detected alleles were imported into the Microsoft Excel program. Next, the number of reads for alleles with identical 30 nucleotides around the predicted cut site were consolidated. Alleles with a frequency of <0.01% were excluded from the analysis. Finally, the relative frequency of the ten most frequent alleles among the consolidated alleles was calculated.

*LNPs for in vitro experiments; preparation, characterization, and concentration*: LNPs were manufactured using microfluidic mixing[52,53]. Stocks of lipids were dissolved in ethanol to obtain a lipid concentration of 1–2 mM. The concentrations of the lipid stocks were carefully evaluated in order to maximize the number of loaded LNPs in order to increase the efficacy. xRNA (mRNA and/or gRNA) was diluted in RNase free 100 mM citrate buffer pH 3.0. The aqueous and ethanol solutions were mixed in a 3:1 volume ratio using a microfluidic apparatus NanoAssemblr (Precision NanoSystems Inc. Vancouver, Canada), at a mixing rate of 12 mL/min. LNPs were dialyzed overnight using Slide-A-Lyzer G2 dialysis cassettes (Thermo Scientific). The size of the LNPs was measured by dynamic light scattering (DLS) and performed on diluted samples in 10 mM phosphate buffer at 25 °C. Data were collected at a scattering angle of 173° and the reported diameter is a mean of three values. Z-average, number, and volume weighted particle size distributions were calculated using a particle refractive index of 1.45 and an absorption of 0.001, and polydispersity (PDI) of the LNPs was determined using a Zetasizer Nano-ZS (Malvern Instruments Ltd). The size (Z-average) of the LNPs prepared was in the range of 74–87 nm with a PDI of 0.14–0.21. The LNPs were concentrated to 100 μg/ml by ultra-spin filtration in Amicon Ultra-4, 30,000 kDa MWCO, centrifugal filter units, and the size was re-measured by DLS in order to confirm the integrity of the LNPs. The encapsulation and concentration of mRNA were determined using the RiboGreen assay (Thermo Scientific) and the encapsulation efficiency for all samples was typically in the range of 92–96%.

*Blood collection and triglyceride determination*: Blood from vena saphena was collected in EDTA-coated tubes, centrifugated (7000 g, 10 min), and plasma was stored at −20 °C. Plasma triglyceride was measured in all mice 1 week before treatment and 3 weeks after LNP transduction using an enzymatic colorimetric method (Cat. A11A01640; ABX Pentra 400; HORIBA Medical, Irvine, CA, USA).

*SDS-PAGE and Western blot analysis*: Cells, isolated tumors, or tissue were lysed in RIPA buffer containing cOmplete Protease Inhibitor Cocktail (Roche, Germany) using a TissueLyser II (25 Hz for 1 min), cell debris was removed by centrifugation at 10,000 rpm for 10 min and protein concentration was determined using the Pierce™ BCA Protein Assay Kit (Thermo Fisher Scientific). Samples were prepared for sodium dodecyl sulfate-polyacrylamide gel electrophoresis (SDS-PAGE) (30 μg loaded per well), separated by SDS-PAGE, transferred onto a PVDF membrane (Invitrogen, the Netherlands), and probed with rabbit pAb CRISPR/ Cas9 (C15310258, Diagenode, Liège, Belgium, 1:5000), rabbit pAb GFP (ab290; Abcam, Cambridge, UK, 1:5000), and rabbit pAb GAPDH (9485; Abcam, Cambridge, UK, 1:10,000) (summarized in Supplementary Table 1). Single lung tumors, control lung tissue, and cell lines derived from GEMMs were stained with rabbit mAb LKB1/Stk11 or mouse mAb p53 (3047 and 2524, Cell Signaling Technology, MA, USA; 1:5000) with mAb vinculin used as loading control (Sigma-Aldrich, MO, USA; 1:10,000). Cell lysates were stained with MCT1 antibody (internal AZ generated 1:1000), rabbit pAb CDK12 antibody (Cell Signaling Technology, MA, USA, #11973, 1:1000), rabbit mAb β-Actin (Cell Signaling Technology, MA, USA D6A8), and rabbit mAb GAPDH (Cell Signaling Technology, 14C10, 1:3000). IRDye 800CW-labeled goat anti-rabbit (1:15,000) and 680RD labeled donkey anti-rabbit (1:15,000) (LI-COR Biosciences, Cambridge,

UK) were used for detection and blots were analyzed with the Odyssey infrared imaging system and software (LI-COR Biosciences). Images quantified using ImageJ software and the method of Thermo Fisher[54]. Quantification is performed by subtracting background signal from signal-detected protein of interest (PI, Cas9/GFP) and normalization control (NC, Actin/GAPDH). A relative NC (rNC) value is derived by dividing each individual NC to the highest signal among the NC in each gel. Finally, all background subtracted PI intensity values are divided by each individually matching rNC.

*BaseScope in situ hybridization assay*: BaseScope assay from ACD was performed, sections were baked, deparaffinized, and dried[28]. Pretreatment 1, 2, and then 3 were then conducted prior to BaseScope probe application (custom probes; I-BA-Mm-Kras-Donor-G12C-1zz-st, I-BA-Mm-Kras-Donor-G12D-1zz-st). The slides were then incubated in reagents AMP0, AMP1, AMP2, AMP3, AMP4, AMP5, and AMP6 with rinsing between each step. Finally, slides were incubated with Fast Red and then counterstained with hematoxylin mounting in VectaMount (Vector labs, Burlingame, CA). Semi-quantitative assessment of the BaseScope staining was performed using the ACD guidelines.

*MRI*: All MRI measurements were performed on a Biospec 9.4T/20 MRI scanner (Bruker BioSpin, Karlsruhe, Germany) equipped with a 400 mT/m actively shielded gradient system with ParaVision (PV5.1) software. For MRI measurement, a 50-mm i.d. quadrature resonator (m2m Imaging, Cleveland, Ohio) was used. Prior to the start of the MRI, animals were weighed and then anesthetized. Anesthesia was induced using 4% isoflurane (Attane vet®, 1000 mg/g, Piramal Healthcare, UK) in oxygen and maintained with 1.5–2.5% isoflurane in an air-oxygen mixture during the imaging session. The mouse was placed supine in a Plexiglas cradle and temperature maintained at ~36.5 °C using a heating pad for the duration of the imaging session. MRI acquisitions were synchronized with the respiratory cycle using a respiratory pad placed under the abdomen of the animal to minimize physiological artefacts (SA Instruments, Stony Brook, NY).

The imaging protocol consisted of three scans. First, an ungated gradient echo localizer scan (TR/TE/α: 175 ms/3.8 ms/20°, number of averages (NA): 2, field of view (FOV): 30 × 30 mm, matrix size: 128 × 128) to localize the lungs. Second, a set of two gated multi-slice coronal scans acquired with a fat suppressed Rapid Acquisition with Relaxation Enhancement (RARE) pulse sequence with the following parameters: TR/TE/RARE factor: 826 ms/9 ms/8, NA: 8, FOV: 30 × 30 mm, in-plane resolution: 117 × 117 µm, number of slices: 9–10, interslice: 1.4 mm and slice thickness: 0.7 mm. The total acquisition time was ~10 min per scan. Slice position of the second RARE scan was interleaved into the interslice gap of the first RARE scan in order to cover the whole thoracic cavity without any slice gaps.

*Segmentation of tumor volume*. The two multi-slice RARE scans were first imported into the software ImageJ® (version 1.52a, NIH, MD, USA) and interleaved to obtain a single 3D image data set of the whole thoracic cavity (matrix size: 256x256x18-20). Tumor nodules were segmented semi-automatically using the image analysis software Analyze 12.0 (Biomedical Imaging Resource, Mayo Clinic, Rochester, MN). Segmentation was performed as follows: (1) lung regions comprising tumor nodules were manually contoured in each slice, (2) tumor nodules in each defined region were then segmented based on histogram thresholding. Volumes of tumor nodules (in mm³) were calculated by multiplying the number of segmented voxels by the voxel volume resolution. Total tumor volume was determined by summing the nodule volumes from each slice. To better evaluate and distinguish pulmonary nodules from normal lung parenchyma, MRI images are represented with inverted grayscale.

*Statistics and reproducibility*: For proof of concept experiments representative results are presented while as for statistical calculations a minimum of $n = 3$ biological independent experiments have been performed. Data visualizations and statistical comparisons between groups were performed using GraphPad Prism (v7.02). ANOVA and, when relevant, Dunnett's post hoc test was performed for correction of multiple hypotheses. $P < 0.05$ was considered to be statistically significant. The level of significance in all graphs is represented as follows: *$P < 0.05$, **$P < 0.01$, ***$P < 0.001$, and ****$P < 0.0001$. Exact $P$ values are presented in the description of the results. Further information on research design is available in the Nature Research Reporting Summary linked to this article.

**Reporting summary**. Further information on research design is available in the Nature Research Reporting Summary linked to this article.

## Data availability

Data supporting the findings of this study are presented within the article and supplementary figures. Additional details and data to support the findings of this study are available from the corresponding authors upon reasonable request. Source data are provided with this paper.

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

## Acknowledgements

We acknowledge Anne Goeppert and Ben Taylor for contributions in developing ODInCas9 cell lines. We thank Mikael Bjursell, Marie Johansson, Johan Johansson, Sara Torstensson, Liselotta Hallengren, Anna Thoren and Lillevi Kärrberg for help with the in vivo experiments. Thanks to Marianna Yanez Arteta who helped with the lipid nanoparticle formulation and Mike Firth for assistance with Amplicon Seq analysis. This project has received funding from the European Union's Horizon 2020 research and innovation programme under the Marie Skłodowska-Curie grant agreement No 765269 (S.W).

## Author contributions

M.M., L.M.M., and M.B-.Y. conceived the project; A.L., M.J.P., H.J., R.N., S.T.B., E.J.D., M.M., and M.B.-Y. designed the study; X.X. designed and built the insulated plasmid backbone, A.L., H.J., S.W., J.B., generated cell lines; H.J., M.M., and M.B-.Y. designed and generated the construct and the mouse model; A.L., M.J.P., H.J., R.N., S.W., E.J., S.T.B., E.J.D., J.B., T.A., M.C., and L.B. performed the experiments; A.L., M.J.P., F.S., A.W.B., B.A., S.T.B., C.P.M., E.H., E.J.D., M.B.-Y., and M.M. contributed to data analysis and interpretation; F.S. and C.J. performed and reported on immunohistochemical studies; A.W.B. performed the MRI acquisition and data analysis; A.T.-G. performed bioinformatics analysis; A.S. provided LNP formulations; A.L., M.J.P., S.T.B., and M.M. wrote the manuscript, S.T.B., L.B., E.J.D., C.P.M., A.W.B., F.S., and M.M. contributed to manuscript revision; and M.B-.Y. and M.M. supervised the study. All authors discussed the results and approved the final manuscript.

## Competing interests

A.L., M.J.P., F.S., C.J., A.W.B., S.T.B., S.W., E.J., C.P.M., E.H., T.A., A.T.-G., J.B., A.S., L.B., A.S., M.C., R.N., M.B-.Y., and M.M. are employees and shareholders of AstraZeneca. E.J.D., H.J., X.X., and L.M.M were employees and shareholders of AstraZeneca. A.L. is fellow and M.J.P., H.J., L.B., and A.T.-G. were past fellows of the AstraZeneca Postdoc program.
