## [Peer Review File · Nature Communications]

Reviewers' Comments:

Reviewer #1:

Remarks to the Author:

The authors developed a novel transgenic ObLiGaRe doxycycline-inducible Cas9 system, which makes the expression of spCas9 strictly regulated in time. Below are specific comments.

Major points:

1. Please display the primer sequence and sgRNA sequence in the form of a table in the supplementary material.
2. In this manuscript, the editing efficiency of two sgRNAs does not exceed 60%, which indicates that the editing efficiency of sgRNAs is not robust. The reason for this result is whether there is no sgRNA screening or the limitation of Cas9 expression or there is a problem with the detection of indels. Please discuss.
3. The author needs to repeat the results in Supplementary Figure 1h, "the HCT116 cell line".
4. It would be better if a description of "ObLiGaRe" was added in the introduction.
5. A side effect assessment of doxycycline is required. How many doses of doxycycline can damage cells or mice?
6. The system cannot achieve perfect spatial control. The occurrence of unexpected tumors in non-target tissues illustrates this problem. Is there any way to solve this problem? Please discuss.
7. When using this system to construct the mouse tumor model, will the generation of non-targeted tumors have an impact on the drug treatment experiment? Please discuss.

Minor points:

1. Please show the full name of "ObLiGaRe" earlier in the manuscript
2. Please confirm the picture of the non-induced group of colon cells in Figure 1d.
3. Please leave a space before the unit, for example, replace "100µg/ml" for "100 µg/ml" in line 422.
4. Please add a scale bar to Supplementary Figure 4B etc.
5. Please check again if the unit in the text is missing, for example, "2C/s" in line 575.
6. Please replace "Fig. 3c,d" for "Fig. 3c" in line 198.
7. Please replace "pERK1/2" for "F480" in Supplementary Figure 4B e.
8. Please replace "Supp. Fig 4B b" for "Supp. Fig 4B a" in line 271.
9. Please check whether the content of the manuscript is consistent with the corresponding figure notes in brackets, for intense, in line 273-307.

Reviewer #2:

Remarks to the Author:

In this article Lundin and colleagues aim to generate a novel system for functional studies using a tightly regulated all-in-one inducible Cas9 system for use in cell lines and mouse models. They highlight advantages over current systems in that the ODInCas9 system is highly sensitive to doxycycline activation and tunable. As the system turns off upon withdrawal of doxycycline it can be repeatedly activated for sequential editing of multiple genes. The authors demonstrate the capability of generating genome edited cells from different tissues of origin by integrating the ODInCas9 transgene into multiple human cancer cell lines as well as human induced pluripotent stem cells (hiPSC). Furthermore, they validate the technology in mice by generating multiple ODInCas9 transgenic mouse models including Kras GEMMs.

The methodology, results and figures are nicely presented and represent a technologic advance. Novelty is somewhat limited, however, by established inducible in vivo CRISPR models (eg Dow et al., Nat Biotechnol 2015; PMID 25690852), and KRAS GEMMs are fairly straightforward to generate by multiple technologies. The particular advantage of their system given its tight

regulation would be to validate in vivo cancer dependencies (eg to perform inducible knockout of a cancer target such as MEK/ERK or other targets in the Kras/Trp53 GEMM).

Specific comments:

- Did the authors perform a western-blot or IHC of p53 or Lkb1 in KRAS GEMMs to validate protein loss?

-The images of immunohistochemistry and well as histological slides in Figures 4 and 5 should contain measure bars (μm).

-Indel graphs are missing error bars

-Please make sure that the manuscript text and the figure legends and footnotes match with the mouse models in order to better understand the information:

According to the manuscript (Line 274): A robust tumor development was observed in all models; KrasG12D;Trp53^{-/-} (Fig. 4b), KrasG12C;Trp53^{-/-} (Supp. Fig 4A b) and KrasG12D;Stk11^{-/-} (Supp. Fig 4B b) each with slightly different attributes (..)

According to the legend of the figures (..):

- Figure 4A Supplementary - ODInCas9 KrasG12D;Stk11^{-/-} NSCLC model
- Figure 4B Supplementary - ODInCas9 KrasG12C;Trp53^{-/-} NSCLC model

There is a mismatch with the figure 4B Supplementary and its footnote, since the model that appears represented is the KrasG12D Trp53^{-/-} instead of the ODInCas9 KrasG12C;Trp53^{-/-}

Reviewer #1 (Remarks to the Author):

The authors developed a novel transgenic ObLiGaRe doxycycline-inducible Cas9 system, which makes the expression of spCas9 strictly regulated in time. Below are specific comments.

Major points:

1. Please display the primer sequence and sgRNA sequence in the form of a table in the supplementary material.

Primer and gRNA sequences were provided in the supplementary material however we failed to provide a reference to look for the information there. We have now updated the text to include a table and a statement linking to this important information. Line 161 now reads; 'Synthetic crRNA sequence information is within Table 1, Supp. Info Table 1.'

2. In this manuscript, the editing efficiency of two sgRNAs does not exceed 60%, which indicates that the editing efficiency of sgRNAs is not robust. The reason for this result is whether there is no sgRNA screening or the limitation of Cas9 expression or there is a problem with the detection of indels. Please discuss.

This reviewer raises some interesting points. Prior to performing proof of principle experiments of ODInCas9 construct editing across the cell lines, no sgRNA screening was performed to optimize editing efficiency. Further, in the proof of principle experiments we used a semi-quantitative mismatch-specific endonuclease activity analysis. Moreover, the experiment was performed using plasmid expression of sgRNA meaning that the editing efficiency is dependent on transfection efficacy and not sgRNA efficiency alone.

To address the detection level of sgRNA Cas9 editing we have performed an additional experiment. The ODInCas9 hiPSC line was transfected with sgRNA and a quantitative NGS analysis performed providing the high resolution editing results presented in Supplementary Figure 1i. These results showed over 85% editing efficiency in hiPSCs.

3. The author needs to repeat the results in Supplementary Figure 1h, "the HCT116 cell line".

We recognize that the original gel image provided in Supplementary Figure 1h from the sgRNA transfection experiment using the ODInCas9 HCT116 contained some unspecific bands which were confusing. To address this we re-ran the experiment to generate a new clone of the HCT116 and in parallel we also generated an ODInCas9 clone of the colorectal adenocarcinoma cell line DLD1. Once the clones were identified T7 assay was performed to show ODInCas9 specific editing in doxycycline stimulated cells transfected with guide RNA. Identical results were achieved for each transfection, further supporting the rapid nature of cell line generation using the ODIn system. The gel image of the HCT116 provided below is now included in Supplementary Figure 1h.

4. It would be better if a description of “ObLiGaRe” was added in the introduction.

To the text at line 95 we have added the phrase ‘Obligate Ligation-Gated Recombination ObLiGaRe’ and at line 99 we have added the following; ‘ObLiGaRe is the first described knock-in method of targeted integration by NHEJ that is using nuclease mediated cleavage of the Donor vector and the genomic target locus followed by endogenous NHEJ mediated ligation. ObLiGaRe allows efficient and directional integration of monomer or ligated multimers of a cassette of interest at a specific locus. The use of NHEJ renders the integration mechanism independent from cell cycle status and the size of the cassette, giving more flexibility respect to Homology Dependent systems of targeted integration.’

5. A side effect assessment of doxycycline is required. How many doses of doxycycline can damage cells or mice?

We agree with the reviewer that doxycycline treatment of mice is known to result in unwanted or harmful effects, as noted in 1988 by Riond¹. The Tet-on system, which became widely used in mice in the 1990s, utilizes a wide range of doxycycline dosing strategies. Mice can be provided doxycycline in food, drinking water, intraperitoneal injection or by oral gavage with the duration of treatment ranging from days to months. We chose to use 2 mg/ml delivered by drinking water as the delivery / dosing method does not require technical skill like gavage dosing. Further, as reported by Hasan in 2001 and as our results demonstrate in Figure 3d and Supp. Fig. 3b, doxycycline induction of the Tet-on system differs across organs². Since we wanted to explore the full expression of Cas9 in the

ODInCas9 mouse, the dose of 2 mg/ml was chosen as this allows difficult organs such as the brain to be ‘activated’.

We have investigated different doses and delivery methods of doxycycline in the ODInCas9 mouse to

explore possible toxicity from doxycycline. Heterozygous ODInCas9 mice (n=4 / group) were dosed by oral gavage (PO) with either, vehicle, 50 mg/kg, 200 mg/kg or 600 mg/kg or were provided with 2 mg/ml doxycycline in drinking water for 4 days. The results above demonstrate that drinking water administration is superior to oral gavage; weight loss was significant on the first day but improved as treatment progressed. Food consumption was only minimally reduced and recovered quickly after cessation of treatment. The 200 mg/kg or 600 mg/kg PO doses resulted in significant weight loss and hepatocellular necrosis with severity increasing with concentration. No changes in liver or GI tract morphology were observed with drinking water administration of doxycycline. We conclude that short term, 2 mg/ml doxycycline in drinking water results in no cellular damage to the mouse whilst ensuring maximum Cas9 expression in all organs. Further, titration of doxycycline dose can be performed if the organ of interest expresses high levels of Cas9 early (@24 hrs) such as the GI tract and pancreas (Supp Fig 3b).

We have added the following text to line 657; 'No significant weight loss, hepatocellular necrosis or changes in liver or GI tract morphology were observed with drinking water administration of doxycycline'.

1. Riond et al., *Vet Hum Toxicol* 1988;30(5):431-443
2. Hasan et al., *Genesis* 2001;29:116-122

6. The system cannot achieve perfect spatial control. The occurrence of unexpected tumors in non-target tissues illustrates this problem. Is there any way to solve this problem? Please discuss.

The reviewer is correct, Intravenous injection is not targeted by definition. Control of tumor induction with this experimental system is achieved by direct delivery of virus to a target tissue. We have added the following text to line 387 of the manuscript text to discuss spatial control of the CRISPR events; 'As expected we observed tumors in non-target tissues by using i.v. injection of virus but tissue specific delivery like oropharyngeal aspiration restricted tumor development to the lungs. Therefore tissue specific delivery is one way to achieve spatial control. An alternative way would be to express the rtTA under a tissue specific promoter as described by Schönig and Bujard[38]. Finally it is possible to achieve spatial control of CRISPR by combining miR expression with anti-CRISPR regulation as described by Wang[39].'

7. When using this system to construct the mouse tumor model, will the generation of non-targeted tumors have an impact on the drug treatment experiment? Please discuss.

When delivering virus to the lung by oral aspiration we have never detected a metastatic tumor outside of the lungs, by MRI or visual assessment of liver, gut, brain, and all major organs at necropsy. Further, i.v. dosing of AAV was explored to assess systemic delivery. This gave 1 tumor in the liver (of a rare phenotype unlikely to be related to Cas9 editing but cannot be truly ruled out) and 1 additional lesion on the nose of the mouse. Use of targeted guide delivery to limit non-specific tumor formation is discussed above. If delivery of a guide with a tissue specific promoter is used in conjunction with selection of an AAV with tissue specific tropism, non-targeted tumors should be minimal.

Minor points:

1. Please show the full name of "ObLiGaRe" earlier in the manuscript

We have added the descriptor to line 94 and the introduction to the ODInCas9 mouse now reads 'Here, we describe the generation a novel transgene, **Obligat** **Ligation-Gated Recombination** ObLiGaRe Doxycycline Inducible Cas9 (ODInCas9), an all in one, universal Tet-On system where a

combination of insulators placed in a modular vector allows tight temporally regulated expression of *Streptococcus pyogenes* Cas9 (SpCas9).'

2. Please confirm the picture of the non-induced group of colon cells in Figure 1d.

As the reviewer has also asked that we redo the sgRNA transfection experiment using the HCT116 line a new fluorescent image has been provided from the newly generated HCT116 clone that has the matching T7 editing results in Supplementary Figure 1h.

3. Please leave a space before the unit, for example, replace "100µg/ml" for "100 µg/ml" in line 422.

We thank the reviewer for proof reading our manuscript. There were unfortunately numerous occasions where the space was missing between the number and the unit of measure. We have corrected this instance and others, they are highlighted in the text by using track changes.

4. Please add a scale bar to Supplementary Figure 4B etc.

We thank the reviewer for highlighting the need to add a scale bar to Figure 4B. Upon addition of scale bars to this figure we noticed they were also lacking from other figures. Scale bars have now been added to Figure 3, Supplementary Figure 3, Figure 4, Supplementary Figure 4A, Supplementary Figure 4B and to Figure 5 and lastly Supplementary Figure 5 and the figure legends adjusted accordingly.

5. Please check again if the unit in the text is missing, for example, "2C/s" in line 575.

After providing additional information to the text this line number has now become line 589 and reads; 'PCR amplified target sequence was heated to 95°C and slowly (2 °C/s) cooled down to 25°C to form heteroduplexes before incubation with mismatch specific endonuclease which cleaves heteroduplexes enabling detection of indel formation.'

6. Please replace "Fig. 3c,d" for "Fig. 3c" in line 198.

This has become line 206 and reads as 'Moreover, the classic mendelian inheritance observed when bred to C57BL/6NCrl mice (Fig. 3c) suggest no background toxicity from the construct.'

7. Please replace "pERK1/2" for "F480" in Supplementary Figure 4B e.

We thank the reviewer for noticing this error. We have now corrected the text above the image in Supp. Fig. 4B e.

8. Please replace "Supp. Fig 4B b" for "Supp. Fig 4B a" in line 271.

We thank the reviewer for highlighting inconsistencies with our referencing to the appropriate part of Figure 4. Since we have now included Western blot data from cell lines generated from each model and single tumors, we have reformatted Figure 4 and have reference to the correct panels in the text. Lines 286-320 now contain yellow highlights to the new Figure 4 and Supp. Fig. 4 parts.

9. Please check whether the content of the manuscript is consistent with the corresponding figure notes in brackets, for intense, in line 273-307.

See above

Reviewer #2 (Remarks to the Author):

In this article Lundin and colleagues aim to generate a novel system for functional studies using a tightly regulated all-in-one inducible Cas9 system for use in cell lines and mouse models. They

highlight advantages over current systems in that the ODInCas9 system is highly sensitive to doxycycline activation and tunable. As the system turns off upon withdrawal of doxycycline it can be repeatedly activated for sequential editing of multiple genes. The authors demonstrate the capability of generating genome edited cells from different tissues of origin by integrating the ODInCas9 transgene into multiple human cancer cell lines as well as human induced pluripotent stem cells (hiPSC). Furthermore, they validate the technology in mice by generating multiple ODInCas9 transgenic mouse models including Kras GEMMs.

The methodology, results and figures are nicely presented and represent a technologic advance. Novelty is somewhat limited, however, by established inducible *in vivo* CRISPR models (eg Dow et al., Nat Biotechnol 2015; PMID 25690852), and KRAS GEMMs are fairly straightforward to generate by multiple technologies. The particular advantage of their system given its tight regulation would be to validate *in vivo* cancer dependencies (eg to perform inducible knockout of a cancer target such as MEK/ERK or other targets in the Kras/Trp53 GEMM).

Specific comments:

1. Did the authors perform a western-blot or IHC of p53 or Lkb1 in KRAS GEMMs to validate protein loss?

To fully address this important point western blot analysis has been performed on both cell lines generated from isolated Kras;Trp53^{-/-} and Kras;Stk11^{-/-} tumors and from individual whole tumors excised directly from lung tissue. The cell lines were derived to allow for 'cleaner' analysis of protein knock out as they are derived from the epithelial fraction of the tumor. This analysis clearly confirms modification of the target genes. We also provide western blots of whole excised tumour fragments where target protein reduction is observed, however expression is not ablated. This is because of the potential for stromal tissue within the tumour to express trp53 and STK11 and therefore resulting in a seeming less efficient reduction in protein in target tumors. For the reduction in trp53 the level of reduction observed associated with generation of adenocarcinoma is similar to that described in other studies in NSCLC GEM models (Ji et al., Nature 2007;448:807-10, Tchaicha et al., Cancer Res 2014;74(14);4676-84). One caveat is that the stage/grade (e.g. adenoma vs adenocarcinoma) of the single tumors is unknown as the tumor were selected visually (~1mm in size) and the whole tumour node lysed for western analysis. The results from these blots are consistent with the work of the Wang laboratory, now cited in the manuscript.

The following has been added to the manuscript text at line 311; Western blot analysis for p53 expression showed that p53 was not detected in 8/10 cell lines derived from the Kras;Trp53^{-/-} co-mutated tumors (Fig. 4c). Importantly, Trp53 cDNA sequencing identified exon 7 deletions in the 2 p53-expressing cell lines (Fig 4c and data not shown), suggesting that they may express a non-functional p53 protein. While we have not confirmed that this is non-function Trp53 it demonstrates that in each line Trp53 was successfully targeted. Trp53 expression was observed in the cell lines derived from tumours where Kras;Stk11^{-/-} were co-targeted demonstrating selective loss of p53 with the appropriate guide RNA. Similarly in the cell lines derived from tumors where Kras;Stk11^{-/-} were modified Stk11 protein was not detected, while STK11 protein was expressed in the cell lines derived from Kras;Trp53^{-/-} tumors. The cell line data confirm effective and selective targeting of the appropriate proteins. The expression of the target genes was also assessed in whole tumor. Reduction in Trp53 protein was also seen in Kras;Trp53^{-/-} targeted tumors relative to normal tissue and Kras;Stk11^{-/-} targeted tumours with a 70 and 75 % reduction in p53 and Stk11 protein relative to normal lung tissue, respectively, confirming the corresponding genotypes in these heterogenous masses of tumour and stromal cells (Fig. 4d). Of note, these results are in line with reports from other NSCLC models [26, 27].

2. The images of immunohistochemistry and well as histological slides in Figures 4 and 5 should contain measure bars (μm).

We thank the reviewer for highlighting the need to add scale bars to our Figures. Scale bars have been added to Figure 3, Supplementary Figure 3, Figure 4, Supplementary Figure 4A, Supplementary Figure 4B and to Figure 5 and lastly Supplementary Figure 5 with the figure legends adjusted accordingly.

3. Indel graphs are missing error bars

As the proof of principle experiments of ODInCas9 editing across cells lines, lines were analysed with a semi-quantitative mismatch-specific endonuclease activity method. We have now performed an additional experiment where the ODInCas9 hiPSC line was transfected with sgRNA and a quantitative NGS analysis generated to provide high editing resolution with statistical power. The data are represented in Supplementary Figure 1i.

4. Please make sure that the manuscript text and the figure legends and footnotes match with the mouse models in order to better understand the information:

According to the manuscript (Line 274): A robust tumor development was observed in all models; KrasG12D;Trp53^{-/-} (Fig. 4b), KrasG12C;Trp53^{-/-} (Supp. Fig 4A b) and KrasG12D;Stk11^{-/-} (Supp. Fig 4B b) each with slightly different attributes (..)

According to the legend of the figures (..):

- Figure 4A Supplementary - ODInCas9 KrasG12D;Stk11^{-/-} NSCLC model
- Figure 4B Supplementary - ODInCas9 KrasG12C;Trp53^{-/-} NSCLC model

There is a mismatch with the figure 4B Supplementary and its footnote, since the model that appears represented is the KrasG12D Trp53^{-/-} instead of the ODInCas9 KrasG12C;Trp53^{-/-}

We thank the reviewer for highlighting the errors we made when referring to our data in both Figure 4 and Supplementary 4A and 4B. We have now added more data to Figure 4 and corrected the referencing to Supp. Fig. 4A and 4B. The corrections are highlighted in the text in yellow. Line 278 now reads as 'A robust tumor development was observed in all models; KrasG12D;Trp53^{-/-} (Fig. 4b), KrasG12D;Stk11^{-/-} (Supp. Fig 4A a) and KrasG12C;Trp53^{-/-} (Supp. Fig 4B a) each with slightly different attributes (Table 1).' Further, we have corrected the footnote in all Figures and changed the title and subtitle in Supp Fig 4B to the correct *Kras* genotype, *Kras* G12C.

Reviewers' Comments:

Reviewer #1:

Remarks to the Author:

No more comments

Reviewer #2:

Remarks to the Author:

The authors have satisfactorily addressed my concerns